# Biomimetic Scaffolds for Tendon Tissue Regeneration

**DOI:** 10.3390/biomimetics8020246

**Published:** 2023-06-09

**Authors:** Lvxing Huang, Le Chen, Hengyi Chen, Manju Wang, Letian Jin, Shenghai Zhou, Lexin Gao, Ruwei Li, Quan Li, Hanchang Wang, Can Zhang, Junjuan Wang

**Affiliations:** 1School of Savaid Stomatology, Hangzhou Medical College, Hangzhou 310000, China; 2School of Basic Medical Sciences and Forensic Medicine, Hangzhou Medical College, Hangzhou 310000, Chinalq953096952@outlook.com (Q.L.); 3School of Pharmacy, Hangzhou Medical College, Hangzhou 310000, China; 4School of Medical Imaging, Hangzhou Medical College, Hangzhou 310000, Chinawhc1234562023@outlook.com (H.W.); 5Department of Biomedical Engineering, College of Biology, Hunan University, Changsha 410082, China

**Keywords:** tissue engineering, tendon regeneration, biomimetic scaffolds, regenerative medicine

## Abstract

Tendon tissue connects muscle to bone and plays crucial roles in stress transfer. Tendon injury remains a significant clinical challenge due to its complicated biological structure and poor self-healing capacity. The treatments for tendon injury have advanced significantly with the development of technology, including the use of sophisticated biomaterials, bioactive growth factors, and numerous stem cells. Among these, biomaterials that the mimic extracellular matrix (ECM) of tendon tissue would provide a resembling microenvironment to improve efficacy in tendon repair and regeneration. In this review, we will begin with a description of the constituents and structural features of tendon tissue, followed by a focus on the available biomimetic scaffolds of natural or synthetic origin for tendon tissue engineering. Finally, we will discuss novel strategies and present challenges in tendon regeneration and repair.

## 1. Introduction

A tendon is a connective tissue with cord-like architecture that attaches muscle to bone. By exerting a traction on the muscle, tendon tissue facilitates the stretching and contraction of the muscle fibers, allowing for the accumulation and subsequent release of adequate energy to drive high-intensity movements involving repetitive loading [1,2,3]. Therefore, tendon tissue plays a vital role in the transmission of stress and the maintenance of joint stability.

The occurrence of tendon injuries/defects caused by trauma, age-related degeneration, or excessive loading of the musculoskeletal system is a common clinical problem, which can result in substantial pain and disability [4]. Tendons consist of abundant collagen; collagen fibers are responsible for some important functions, including providing structural support, regulating cell behaviors, modulating inflammation, and guiding mechanical transfer under different loads [5,6,7]. Tendon injuries are typically accompanied by collagen fiber bundle fracture and glycosaminoglycan accumulation, which lead to disarrangement of the tissue structure and inferior mechanical properties [3]. Unfortunately, due to the paucity of blood supply and cellular components in tendons, natural healing of tendon tissue is extremely inefficient [8,9]. Hence, tendon repair has become a formidable challenge in clinical environments.

Currently, the conventional clinical treatments recommended for tendon injuries include immobilization [10], physiochemical therapy [11], and surgical suturing [12]. For acute extensor tendon injury, physico-chemical treatments such as corticosteroids, nonsteroidal anti-inflammatory drugs [13], extracorporeal shock waves [14], centrifugal exercise [15], and low-energy lasers are commonly employed to promptly alleviate tendon pain [16]. However, the risk of recurrence is relatively high, with a prolonged treatment period, and the fundamental issue of tendon dysfunction remains unresolved. In chronic tendon injury, surgical excision of fibrous adhesions and suturing of torn tendon tissue is performed to restore peripheral circulation and induce tenocytes that restart the protein secretion process. Although suture surgery provides temporary restoration of tendon continuity, it fails to retain the structural integrity and mechanical strength, and there is a risk of retearing after surgery. The retear rate for small (<1 cm) and medium (1–3 cm) tears of tendons is statistically 26%, and up to 94% for large (3–5 cm) and huge (>5 cm) tears [17]. Furthermore, surgical intervention for chronic tendon injury is highly invasive and may result in postoperative complications such as wound infections, erythema, and inflammatory reactions, leading to complex issues such as peritendinous fibrosis, tendon sheath collapse, and atrophy [18]. Currently, there is indeed need to explore novel treatment modalities that are both efficient and minimally invasive to effectively address the issue of tendon injury.

With the thriving development of tissue engineering, many biomaterials have been demonstrated to promote the regeneration of specific tissues by direct stem cell differentiation, thus providing new insights into the treatment of tendon injuries [19]. Tissue-engineered tendons possess some advantages in comparison to conventional tendon repair, such as perfect morphological restoration, unrestricted material source, excellent biocompatibility, and absence of immune reactions [20]. In this review, we will begin with a description of tendon structure and challenges faced in tendon repair. Subsequently, the application and prospects of tissue engineering in tendon repair are reviewed, before we narrow our focus to introducing the scaffold for tendon repair. Finally, the present challenges and future development directions will be summarized.

### 1.1. Composition and Structure of Tendon Tissue

Microscopically, tendon tissue is made up of scattered spindle-shaped tenocytes and extracellular matrix (ECM) formed by their production of collagen, elastin, proteoglycans, and trace inorganic elements, keeping the interior environment of the tendon tissue at a stable state [21]. Tenocytes are fibroblast-like cells interspersed into collagen fibers or intima, accounting for 5% of the mass of tendon tissue [22]. In contrast, the ECM of tendon tissue occupies 60–85% of its dry mass [23]. Collagen is the main component of tendon ECM, which occupies 65–70% of the shaped portion. Among these, 97–98% is collagen type I, with a few other types of collagen (2–3%), including types III, IV, V, and X [24,25]. It is noteworthy that type III collagen mediates the production of type I collagen, which regulates the diameter and size of collagen fibers [26]. Similarly, type V collagen also has a physiological function in the self-assembly and matrix formation of type I collagen [27]. Except collagen, elastin is the most abundant structural protein in ECM, which improves the tensile capacity and viscoelasticity of tendons after polymerization with type I collagen [2,28,29]. Besides collagens, tendons also contain proteoglycans in small quantities. Proteoglycans consist of a core protein and a glycosaminoglycan (GAG) side chain covalently attached to it. Core proteins are responsible for the early fibril assembly in tendon development by specifically recognizing collagen interaction sites and binding to them. Different core proteins could also be intermolecularly crosslinked via GAG chains that form interfibrillar bridges for load transfer in discontinuous collagen fibers [30]. Overall, the matrix components co-construct the tendon microenvironment to be responsible for the nonlinear viscoelastic and anisotropic behaviors. The relative mechanical properties between collagen fibers and ECM determine the fiber extension and shearing to the total material stretch [31].

Structurally, tendon tissue has a complex hierarchical fibrillar arrangement (Figure 1A) [32]. Procollagen is the basic structural unit of collagen, which is a stable structure formed by three polypeptide chains with a characteristic amino acid sequence repeat unit [-(GlycineXY)n-] coiled around each other by interchain hydrogen bonds [33] with a collagen triple helix of about 1–2 nm in diameter and 300 nm in length after protease shearing to remove the untwisted C- and N-prepeptides (Figure 1B) [34,35]. The procollagen molecules are covalently crosslinked by adjacent cysteines to form collagen fibrils with diameters of 50–200 nm and lengths of 150 nm to several microns in an orderly stepwise arrangement outside the cell [36,37]. Collagen fibrils are bonded with a small amount of mucilage (proteoglycans and glycoproteins) to form collagen fibers about 0.5–20 μm in diameter and 10 mm in length [34]. Collagen fibers are further assembled into the largest subunit of tendon, the fascicle, which varies from 150 to 500 μm in diameter. The fascicles are eventually wrapped in a dense connective tissue sheath to form the tendon tissue [21]. The loose connective tissue membrane that covers the surface of the tendon is the peritendinous membrane, which is an essential channel for vascular nutrition of the tendon for material exchange [37]. Moreover, the double layer of closed synovial tissue covering the joints and other high-frequency activity areas (e.g., wrist, ankle, etc.) is the tendon sheath, which provides effective protection and lubrication, and reduce the mechanical friction between the tendon cavities. The surface of collagen fibers has alternating light and dark periodic transverse patterns with a large number of fiber branches interwoven into a network, which maintains the mechanical structural stability of collagen fibers to withstand the strength under tensile loads driven at different moments [38]. Histologically, collagen fibers exhibit a wavy shape (Figure 1C) [39], which protects them from damage by aligning and stretching to absorb and disperse the stresses applied when subjected to tensile loads [40].

In essence, the intricate architecture of tendons imparts formidable biological pliancy and mechanical strength. However, this complex structure also constitutes a critical impediment to the treatment of tendon injuries.

### 1.2. Tendon Healing: Repair and Challenge

Based on the Global Burden of Disease (GBD), approximately 1.71 billion people worldwide are affected by musculoskeletal disorders, of which tendon-related disorders account for at least 40% to 50% [41]. Typical sites of tendon pathology include rotator cuff tendons, the anterior cruciate ligament (ACL) of the knee, and the Achilles tendon. As early as the 20th century, 16% of the general population were found to suffer from rotator-cuff-related tendon disorders [42], and their incidence rates increase gradually with age. In athletic populations, Achilles tendinopathy alone accounts for 6.2–9.5% of all injuries [43]. The annual cost of medical intervention required to treat extensor tendon rupture in the United States is estimated to be USD 307 million [44]. The economic burden of treating tendon injuries is particularly noteworthy for professional athletes. In the NBA, ACL injuries incurred an economic loss of USD 99 million between 2000 and 2015, with an average cost of USD 2.9 million for rehabilitation per player [45].

The causes of tendon injuries are varied, with individual factors including age, gender, genetics, weight, and underlying disease. Extrinsic factors include exercise, physical load, work, etc. [32]. Tendon healing can be divided into three main phases: inflammatory, proliferative, and remodeling (Figure 2) [46].

#### 1.2.1. Inflammatory Phase

The inflammatory phase begins immediately after injury and lasts for one to two weeks. During this period proinflammatory cytokines predominate and inflammatory cells such as neutrophils, monocytes, and macrophages are recruited from the peripheral blood to the wound site. Briefly, after injury, neutrophils first arrive to trigger an immune response, which peaks about 1 day after damage. Subsequently, monocyte-derived macrophages are activated to remove the necrotic material through phagocytosis [11,47,48]. During this phase, macrophages mainly activation type M1 [49] and secrete inflammatory factors, such as IL-6 and IL-1β. In the meantime, fibroblasts gather at the wounded site to participate in the tendon healing process and form the neocellular matrix. In addition, blood clots form in the damaged site [50]. Platelets and clot cells release transforming growth factor-β (TGF-β), insulin-like growth factor-1 (IGF-1), and platelet-derived growth factor (PDGF) to promote the formation of vascular networks and provide blood supply for fibrous tissue [48,51].

#### 1.2.2. Proliferative Phase

The proliferative phase begins two days after the injury and is characterized by abundant deposition of ECM, increased cell numbers, and the formation of a fibrous scar [50]. During the proliferative phase, TGF-β is released from macrophages, endothelial cells, and epithelial cells, and macrophages gradually transform into the M2 phenotype, which plays an important role in suppressing inflammation and eliminating scarring at a later stage [52,53]. TGF-β is also responsible for regulating protease activity, stimulating collagen production, and the subsequent recruitment of tendon stem/progenitor cells (TSPCs). TSPCs at the wounded site will proliferate and differentiate into tenocytes to promote tendon repair. After that, fibroblasts and tenocytes aggregate to the injured area and deposit collagen, fibronectin, proteoglycan, chondroitin sulphate, and dermatan sulphate [54]. However, at this phage, the ECM remains in a disorganized state.

#### 1.2.3. Remodeling Phase

After about 6 weeks, the remodeling phage starts. This phase is characterized by tissue contouring and maturation, which lasts 1 to 2 years. Although the number of cells and total ECM synthesis decreases, there is a transition from collagen type III to type I and the synthesis of type I collagen increases as the diameters of the deposited collagen become larger [55]. The cells redifferentiate into elongated and aligned mature tenocytes, longitudinally, to improve the mechanical strength of the regenerated tissue. After approximately 10 weeks, the collagen fibers begin to intertwine and the bond between collagen fibers increases the stiffness and tensile strength of the repaired tendon [56]. In addition, tenocytes express α-smooth muscle actin (αSMA), which plays an important role in the remodeling phase [57]. Matrix contraction occurs during this phase to close the gap within the injured tendon, which also leads to the formation of a fibrous scar, while enhancing the stability of the injury site [58]. Additionally, the metabolism of tenocytes and tendon vessels is significantly weaker compared to the first two stages [59].

Overall, the complex multiunit hierarchy aligns the fiber bundles with the long axis of the tendon and affords the tendon’s tensile strength, which in turn leads to the difficulty of healing. Thus, it is of vital significance to develop biomimetic strategies for structure reconstruction to promote tendon repair.

## 2. The Potential of Tendon Tissue Engineering for Tendon Repair

Tissue engineering constitutes a significant biomimetic strategy that offers a novel paradigm for the restoration of tendon tissue. In general, the fundamental procedure involves isolating seed cells from tissue using enzymes or other methods, and expanding them through cultivation in vitro. Subsequently, the expanded cells, growth factors, and biomaterial scaffolds with good biocompatibility are mixed in specific proportions. This results in the attachment of cells to the scaffolds, forming a cell-material composite. Finally, the composite is implanted into the damaged tissue or organ in vivo. Over time, the biodegradable biomaterial gradually degrades and is absorbed by the body, and the implanted cells continue to proliferate and secrete ECM in vivo, eventually repairing the damaged tissues. In recent years, numerous studies have demonstrated that tendon tissue engineering based on biomimetic scaffolds has yielded promising outcomes by reconstructing tissue structure and function for tendon repair. Next, we will briefly summarize the seed cells, growth factors, and scaffolds for tendon tissue engineering.

### 2.1. Seed Cells

Tenocytes, fibroblasts, induced pluripotent stem cells (iPSCs), embryonic stem cells (ESCs), bone marrow mesenchymal stem cells (BMSCs), adipose stem cells (ADSCs), and TSPCs are the seed cells mainly used in tendon tissue engineering. We summarize the advantages and disadvantages of the seed cells in Table 1.

#### 2.1.1. Tenocytes

Tenocytes are specialized cells that reside in tendon tissue. The metabolism and proliferation of mature tenocytes are quite slow. Tendon injury stimulates tenocyte proliferation, synthesis, and the secretion of collagen, which directly repairs the injured tendon [60,61]. Tenocytes were originally used as seed cells for tendon tissue engineering. For example, as early as 1994, Cao et al. constructed a tendon-like tissue that resembled a normal tendon in terms of histology and biomechanics by using tenocytes in a nude mouse model [80]. Given the immune rejection of the allogeneic tendon, they subsequently demonstrated the feasibility with autologous tenocytes in a hen model [81]. However, as somatic cells, tenocytes showed marked changes in their protein expression profile as well as loss of the tenocyte phenotype after expansion in vitro [62]. Although, some studies reported that the proliferation capacity of tenocytes is improved by applying growth factors in the tendon healing process. For example, IGF-1 regulated collagen synthesis and tenocytes proliferation by the activation of PI3K/protein kinase B and ERK pathways [82]. Substantial studies are still required to discover more growth factors to promote tenocyte functions.

#### 2.1.2. Fibroblasts

Both skin fibroblasts and tenocytes originate from the mesoderm, which belongs to the fibroblast lineage. There is an abundant amount of fibroblasts since skin is the largest organ in the human body [63,64]. Liu et al. previously fabricated a composite scaffold with porcine fibroblasts or tenocytes seeded on a polyhydroxy acetic acid (PGA) scaffold, then implanted them into a porcine flexor superficial tendon defect model. It was found that the maximum stress of repaired tendon tissue in the fibroblast and tenocyte groups was 74% and 76% when compared to normal tendon, suggesting that skin fibroblasts are expected to substitute for tenocytes in tendon tissue engineering [83]. Given that tendon tissue is constantly exposed to anisotropic dynamic shear forces for a sustained mechanical loading environment, the researchers modified the aforementioned skin fibroblast-PGA scaffold into a U-shaped spring morphology, which helped to significantly increase in collagen fiber diameter and mechanical strength of repaired tendons compared to the nonmechanically loaded group. Moreover, both skin fibroblasts and tenocytes showed a narrow morphology with a highly similar phenotype [63]. This morphology was suggested to be associated with TGF-β1-mediated contraction of the cytoskeleton in the RhoA/ROCK signaling pathway and mechanical transduction pathway through further study of the topological morphology and cellular interaction mechanisms. In short, TGF-β1 induces the transdifferentiation of narrow fibroblasts into tenocytes to promote tendon repair [84]. Furthermore, fibroblasts are the main effective cells in wound healing, tissue remodeling, and fibrotic scar formation. However, excessive fibroblast multiplication inevitably leads to scar formation [65].

#### 2.1.3. BMSCs

BMSCs are a subpopulation of nonhematopoietic cells primarily found in bone marrow (BM). As multipotent stem cells, they can differentiate into various tissues under specific conditions, such as bone, cartilage, adipose, tendon, etc. [72]. BMSCs are the most widely evaluated cell type in tissue engineering as they can be easily obtained from a BM aspirate and amplified in vitro before transplantation [73]. Attempts to commit BMSCs to the tenogenic lineage had been explored. Young et al. made composite tissue prostheses by using autologous BMSCs and gels, and demonstrated for the first time that BMSCs could promote collagen alignment and enhance tendon quality and mechanical properties in a Achilles tendon defect model [85]. It has been suggested that BMSCs have the potential to secrete collagen type I in vitro [86]. Wu et al. reported that, after BMSCs were subjected to periodic uniaxial stretch stimulation in vitro, they generated large amounts of collagen type I [87]. Recently, an increasing number of studies have demonstrated the significant role of BMSC-mediated paracrine effects in the repair of injured tissues. For example, Huang et al. showed that BMSC exosomes increased the maximum fracture load and stiffness of the regenerating rotator cuff in rats and promoted healing of the tendon–bone joint surface. Furthermore, BMSC exosomes could inhibit U937 cells type I polarization in vitro, thus preventing them from secreting proinflammatory factors and inflammatory responses [88]. Although the treated BMSCs adopted a tendon-like cell phenotype in vitro, its tenogenic differentiation efficiency was less than satisfactory [74], as was their limited in vitro amplification ability and tendency to accumulate genetic variation which restricted their utilization [72].

#### 2.1.4. ESCs/iPSCs

ESCs are totipotent stem cells, which have unlimited proliferation capacity and the potential to differentiate into different cell types [66,67]. It has been found that collagen spatial sequences can rapidly rearrange and repair the damaged area in a fetal tendon injury model, but not in adults [89]. Motivated by this, Chen et al. found that MSCs induced from human ESCs (ESC-MSCs) could promote tendon repair by secreting growth factors including TGF-β3, growth differentiation factor 5 (GDF5), and bone morphogenetic protein 2 (BMP2). Moreover, the gene expressions of collagen type I, type III, and eyes absent homolog 2 (eya2) were remarkably enhanced in patellar tendon tissue treated with human ESC-MSCs, indicating the activation of endogenous repair pathways in tendon tissue [90]. Then, they structured human ESC-MSCs into tissue-engineered tendon tissue; the results of cell labeling and ECM expression showed that human ESC-MSCs not only promoted tendon regeneration but also had an environmental regulatory effect in situ [91]. Although ESCs are known as the most powerful totipotent stem cells, human embryonic tenocyte cell lines have been poorly studied due to ethical issues. Moreover, ESCs have a tendency to form teratomas [68], thus the safety and functionality of their derived cell transplants remain to be investigated in depth.

iPSCs reprogrammed from terminally differentiated somatic cells into ES-like pluripotent stem cells were initially induced by introducing specific transcription factors through gene transfection techniques, thus they present no ethical issues [69]. iPSCs have also been explored as seed cells for tendon regeneration. For example, Zhang et al. previously induced differentiation of human iPSC-MSCs to the teno-lineage by activating mechanical signaling pathways through a stepwise physical substrate change strategy, which obviously enhanced tendon structure and mechanical properties in the Achilles tendon defect model [92]. Tsutsumi et al. applied a mechanical stretch culture system to investigate the effects of Mohawk (Mkx) transfected iPSC-MSCs on tendon repair. They found that this culture system could recruit a great number of cells secreting collagen in the damaged area and promoting tendon regeneration in a mouse Achilles tendon rupture model [93]. In addition, Bavin et al. found that growth factors have a surprising effect on the physiological behavior and biological functions of iPSCs [94]. TGF-β3 promoted the expression of scleraxis (SCX), elastin (ELN), and Tenascin-C (TNC) to a degree [95], which in turn reduced scar formation through paracrine effects [96]. The developmental trajectory of iPSC-derived tenocytes was analyzed utilizing single-cell RNA sequencing, which confirmed the credibility of the theory above on the paracrine effect [97]. Nevertheless, the issues of immunogenicity, potential tumorigenicity [70], and epigenetic variation [71] of iPSCs are considered to be the major pitfalls for clinical applications, which need to be clarified and confirmed by more clinical trials.

#### 2.1.5. ADSCs

ADSCs are a kind of adult stem cells present in adipose tissue and have the potential to differentiate into multiple lineage tissues. They have been intensively studied for the treatment of tendon injuries because of several advantages, such as having an abundant source, being obtained from low invasive procedure, providing a larger number cells in comparison to BMSCs, and having no ethical controversy [75]. For example, the injection of ADSCs for the treatment of collagenase-induced superficial flexor tendinitis in equine forelimbs could improve the mechanical properties of tendon tissue. Franklin et al. treated acute tendon and chronic tendon–bone interface injuries with tail vein injection of ADSCs and showed promising results [98]. However, there are still many issues to be addressed, including the controlled directional differentiation of ADSCs into tenocytes, determination of optimal culture conditions, cell inoculation density, avoidance of ectopic bone formation [99], etc.

#### 2.1.6. TSPCs

TSPCs exhibit similar characteristics to BMSCs, while their capability for tenogenesis is unparalleled by other seed cells, which have attracted a surge of research since discovery [76,77]. For example, Komatsu et al. transplanted TSPC-derived cell sheets into a rat tendon injury model; the histological properties and collagen content of the repaired tendons were significantly improved, indicating that TSPC-derived cell sheets have tremendous potential for the treatment of tendon injury [100]. However, the relationship between TSPCs and tendon regeneration is poorly characterized. Yin et al. identified a subpopulation of nestin^+^ TSPCs utilizing single-cell genetic analysis, and revealing that characteristic molecular markers have a critical role in maintaining the phenotype and differentiation decisions of TSPCs [101]. To further explore the potential repair mechanisms of TSPCs, Zhang et al. first attempted to apply epigenetic small molecules to promote tendon regeneration. It was found that the inhibitor of histone deacetylase (HDAC), Trichostatin A (TSA), and the synergistic effect of topographical cues together directed the differentiation of TSPCs toward the tendon lineage, which provided a novel idea for the repair of defective Achilles tendons [102].

Notably, the biological properties (self-renewal, migration, proliferation, differentiation capacity, etc.) and cellular phenotype of TSPCs are continuously lost with the onset and progression of aging. This results in the disruption of homeostasis and diminished endogenous repair capacity of the tendon endotrophic environment [103], which is likely connected to the JAK-STAT signaling pathway [104]. Rui and his team found that overexpression of Aquaporin 1 significantly inhibited the expression of JAK-STAT target kinases and the phosphorylation of JAK2 and STAT3 [105]. The inhibition of Wnt5a-attenuated senescence, senescence cell polarity changed, and the expression of senescence-associated secretory phenotype (SASP) genes were found in TSPCs. Although TSPCs have shown great potential for tendon tissue engineering, it is hard to obtain TSPCs with a high level of purification with the available isolation techniques [78]. In addition, the source and quantity of TSPCs are highly limited [79], yet it is difficult to avoid replicative senescence during ex vivo expansion. Overall, there is still a long way to go before the large-scale clinical application of TSPCs for tendon repair.

### 2.2. Growth Factors

Growth factors are important regulators of cell survival and function. Tendon injury stimulates the production of a variety of growth factors at multiple stages, especially in the early stages of healing. Growth factors can enhance the biological function of cells, which directly stimulate cell growth and regulate ECM secretion. Meanwhile, increased growth factors can further activate the endogenous stem cells to enhance protein expression including collagen type I and III, thus promoting the healing of tendon tissue. In recent years, a large number of clinical studies have been conducted to investigate the role of the growth factors in tendon healing. Basic fibroblast growth factor (bFGF) [106,107,108], TGF-β [84,90,95,96,106,109], BMP [90,110,111], IGF-1 [82], and growth/differentiation factor (GDF) [90,112,113] are the well-known growth factors for tendon repair. Table 2 briefly summarizes the role of the factors in tendon healing.

### 2.3. Scaffolds

Scaffold is an essential component of tissue engineering, which provides mechanical stability and 3D structure for the growth of regenerative tissue and attachment of biologically active molecules such as growth factors. To develop a viable tissue for tissue replacement, an engineered substitute should mimic the dynamic ECM microenvironment of native tissue [114,115,116,117]. Different materials have unique structural and physicochemical properties. The biological properties of bionic scaffolds, the flexibility and stiffness of the modified material, the surface topography to modulate cell behavior, and tissue regeneration are dependent on the physicochemical composition of the material [118].

The ideal biomaterial scaffolds for tendon tissue engineering are expected to possess the following characteristics: (1) Excellent mechanical properties to provide sufficient strength and stiffness of the tendon to withstand the stress and tensile forces generated by the surrounding environment [119,120]; (2) Favorable biological functionality to support cell adhesion, growth, proliferation, and differentiation to facilitate matrix secretion and tendon tissue development [121,122]; (3) Strong biodegradability and resorption rate to match the cell growth rate of the repaired tissue [123,124,125]; (4) Low immunogenicity and satisfying biocompatibility with the host both pre- and postdegradation [126,127]; (5) Excellent processability that enables fabrication into intricate structures and shapes, such as knitting, weaving, and electrospinning [128].

### 2.4. Biomimetic Strategies for Tendon Repair

An appropriate microenvironment maintains the survival and normal physiological functions of tenocytes, as well as the production and alignment of collagen fibers [16]. Therefore, the biomimetic strategies for tendon repair aim to better mimic the tendon microenvironment, including biological signals, components and structures of ECM, and cell-matrix interactions, which ultimately achieve biomimetic replication in structure and function, and accelerate tendon repair. For example, it is reasonable to coculture stem cells that can secrete collagen proteins and other key components, along with the biological scaffold [129]. In addition, the introduction of bioactive molecules, such as growth factors and matrix proteins, simulates biological signals within the tendon microenvironment [46]. Mechanical stimuli, such as stretching and compression, can also be applied to construct the mechanical environment of the tendon to accelerate the oriented arrangement of cells and synthesis of matrix proteins within the scaffold [130].

In conclusion, the biomimetic scaffolds with ECM-like properties using different biomaterials mediate cellular attachment and endow them with biological functions with a specific spectrum of differentiation, confirming its promising clinical application in the field.

## 3. The Technologies for Tissue Engineering Scaffolds

Various fabrication methods have been developed to fabricate specific scaffolds; the most widely used methods including 3D bioprinting, wet-spinning, and electrospinning. These manufacturing strategies exhibit distinct merits and drawbacks (Table 3) that can be chosen based on the specific requirements of the application.

### 3.1. 3D Bioprinting

3D bioprinting is an emerging tissue engineering technology that operates on the principle of utilizing 3D printing techniques to layer-by-layer deposit bioinks containing cellular components, biomolecules, and other biological materials in accordance with a predetermined 3D model. This process enables the creation of intricate biological structures, while concurrently providing cells with conducive growth environments and structural support, thus facilitating cellular proliferation and the regeneration of damaged tissues. Jiang et al. utilized 3D printing technology in conjunction with a cell-laden composite hydrogel of collagen-fibrinogen to fabricate multilayer scaffolds using polylactic-co-glycolic acid (PLGA) as the printing material. The study demonstrated the feasibility of utilizing 3D-printed multilayer scaffolds for rotator cuff tendon regeneration, which improved mechanical properties and had excellent biocompatibility [158]. Among various types of printing techniques, inkjet printing and extrusion-based bioprinting are commonly employed in the field of bioprinting (Figure 3) [139].

In inkjet printing, it is necessary to sequentially and selectively deposit bioink onto a building platform until the desired structure is formed [131]. There are primarily two types of inkjet printing systems: Continuous Inkjet (CIJ) and Drop-on-Demand (DOD) inkjet printing. The latter can further be classified into thermal, piezoelectric, and electrostatic inkjet bioprinting [132,133]. The working principle of CIJ involves the ejection of ink through tiny nozzles; at the nozzle outlet, the ink is divided into small droplets through high-frequency oscillation. Only the required droplets are printed onto the target surface, while the remaining droplets are recaptured and reused. The advantages of CIJ printing systems include high-speed printing, continuous printing, and compatibility with different types of inks. However, due to the oscillation and spray during droplet formation, CIJ printing exhibits relatively lower resolution and limited precision in droplet placement. Additionally, there is a certain risk of contamination during the recycling and reuse of droplets [134]. DOD operates by forming and jetting individual droplets in the nozzle according to the printing requirements. It offers high resolution, precise droplet placement, and suitability for printing complex structures and details. Moreover, DOD technology is able to utilize various types of bioinks and cell suspensions, making it widely applicable in the field of bioprinting [132]. Wu et al. employed a novel technique known as electrohydrodynamic jetting (E-jetting) to fabricate 3D tendon scaffolds with high porosity and directed micron-scale fibers. The E-jetted scaffolds consisted of tubular multilayered bundles of micron-scale fibers, exhibiting longitudinal connectivity and geometric heterogeneity along the scaffold. The fiber diameter, stacking pattern, and fiber spacing influenced the structural stability of the scaffold, with scaffolds incorporating thicker fibers as the supporting layer achieving enhanced mechanical strength. Compared to conventional electrospun scaffolds, tenocytes cultured on the E-jetted scaffolds demonstrated significantly increased cellular metabolic activity and promoted the expression of type I collagen. E-jetting was explored for the first time as a novel scaffold approach in tendon tissue engineering, offering a 3D fiber scaffold that facilitates ordered tissue reconstruction and potential tendon repair [135]. However, inkjet printing faces limitations in printing high-viscosity materials and high-density cells due to the relatively low nozzle driving pressure [136]. Moreover, low-viscosity materials result in reduced structural integrity of the printed constructs, which do not meet the requirements for subsequent in vitro culture and transplantation [137]. The viscosity factor narrows down the range of applicable bioink materials. Furthermore, during the inkjet printing process, there is a risk of mechanical or thermal damage to the cells, which further restricts the application of inkjet printing technology [138].

In extrusion-based bioprinting, a bioprinter extrudes bioink (a mixture of biological materials and cells) through a fine nozzle or syringe at a controlled speed and pressure. By manipulating the printing parameters and the trajectory of the nozzle, the desired tissue structure can be built layer by layer on the printing platform. This technology does not involve heating processes, allowing for convenient incorporation of cells and bioactive agents. With the continuous deposition of bioink, this technique offers excellent structural integrity [140]. Compared to inkjet printing, extrusion-based bioprinting enables the continuous flow of bioink, simplifying the operation and offering a wider range of choices for bioink materials. However, there are limitations to extrusion-based bioprinting: (1) It can only extrude high-viscosity materials to maintain the desired fibrous structure after deposition. (2) When extruding through microscale nozzles, there is a potential for pressure drop, which may lead to potential apoptosis of cells during and after the printing process. (3) The resolution of current extrusion-based printing technologies is approximately 200 μm, which is lower compared to inkjet and laser printing technologies [139,159]. Toprakhisar et al. utilized decellularized matrix (DECM) hydrogel bioink derived from bovine Achilles tendon tissue to perform bioprinting without the need for support structures or crosslinking agents. They employed a custom-made printer based on an aspiration and extrusion system. The resulting scaffolds exhibited good biocompatibility and mechanical properties lower than those of natural tendons. The bioink demonstrated rapid gelation properties, transforming into a stable hydrogel under physiological conditions, without exerting any toxic effects on encapsulated cells [141]. However, further improvements are necessary to enhance the performance of the bioink in order to obtain scaffolds with sufficient properties.

### 3.2. Wet-Spinning

Fiber spinning represents a prominent and interdisciplinary application within the realm of modern polymer processing, incorporating techniques such as melt-spinning, wet-spinning, and dry-spinning. Although melt-spinning materials exhibit favorable mechanical properties and biocompatibility, the resultant fiber bundles encounter challenges including nonuniform distribution, inconsistent weight, and demanding equipment requirements, which hinder their suitability for tendon repair applications and contribute to their gradual phasing out [160,161,162]. Wet-spinning and dry-spinning are differentiated based on the use of solvents during the spinning process. Dry-spinning demonstrates impressive mechanical performance. However, its rudimentary and rough fabrication process leads to fiber products lacking in bioactivity and the crucial growth factors necessary for effective tendon repair. Additionally, the coarse surface of fibers produced via this method hampers cell adhesion and growth. In contrast, wet-spinning employs solvents capable of dissolving biomacromolecules while retaining their hydration, yielding tendon-like samples with enhanced elasticity [142]. Of paramount importance is its inherent capacity for deliberate manipulation and fine-tuning of fiber diameter, porosity, and pore size [143]. Hence, wet-spinning has emerged as the principal modality for refining tendon repair approaches.

Wet-spinning is a technique characterized by the stretching of dissolved polymer substances into fibers within the diameter range of tens to hundreds of micrometers [163]. In the wet-spinning process, the polymer substance is initially dissolved in a suitable solvent, followed by the passage of the solution through a rotating centrifuge or a series of shear-inducing devices, ensuring a uniform flow state at each junction. Subsequently, the polymer substance is expelled and undergoes stretching and solidification, resulting in the formation of fibers endowed with both strength and elasticity (Figure 4) [164]. The underlying principle hinges on the dissolution of polymer substances, the compelled extrusion of the solution from a designated mold, and the prompt solidification into fibers facilitated by the influence of centrifugal force or shear force [163].

Initially, Nowotny et al. utilized the wet-spinning technique to fabricate chitosan fiber scaffolds for tendon regeneration [165]. Similarly, Rinoldi et al. employed wet-spinning technology to produce oriented hydrogel yarns loaded with MSCs. The results revealed a highly aligned arrangement of MSCs along the fiber axis and an augmented expression of specific tendon-related matrix proteins. However, due to relatively low material stability, these single-layer scaffolds exhibited a reduction in mechanical performance during subsequent experimental procedures, thus failing to meet the required performance criteria of natural tendons [166]. Consequently, the concept of multilayer scaffolds was introduced for tendon repair. Lu et al. employed the wet-spinning technique and utilized the crosslinking of 1-ethyl-3-(3-dimethylaminopropyl) carbodiimide hydrochloride and N-hydroxysuccinimide. This approach facilitated cell adhesion on the aligned collagen fibers and yielded a highly oriented distribution with favorable cellular morphology, which contributed to the refinement of the fiber surface roughness and promoted the growth of tenocytes [142].

While wet-spinning offers the aforementioned benefits, it also presents certain limitations. The use of wet-spinning to fabricate fibers results in significant variations in fiber diameter with poor size stability. This variability may impact the mechanical performance and biocompatibility of the scaffolds [144,145,146]. Furthermore, wet-spinning is constrained by fiber properties, as the performance of fibers depends on the material and preparation conditions. Careful control of these parameters is necessary to achieve the desired performance [147]. Moreover, compared to electrospinning, this technique is limited to the production of micron-scale fibers, which restricts its applicability in certain scenarios [148].

### 3.3. Electrospinning

The utilization of electrospinning technology seamlessly addresses the inadequacies associated with the traditional wet-spinning method in terms of the formation of fiber diameter irregularities. This process enables the fabrication of individual fibers with diameters ranging from 10 nm to 10 μm [149], spanning the levels of micrometers, submicrometers, and even nanometers. The underlying principle entails the utilization of a high-voltage electric field to impart a substantial electrostatic charge to the target material (polymer solution or molten state). Subsequently, the polymer droplet surpasses surface tension and gives rise to a refined jet, accompanied by the progressive evaporation or solidification of the solvent during the jetting procedure. Ultimately, the stream is emitted from a metallic needle or nozzle in the form of a Taylor cone and gathers onto a collector, culminating in the formation of a nonwoven fabric-like structure comprising fibers at the nanoscale level (Figure 5) [149,167]. Scaffolds obtained through the utilization of electrospinning technology typically exhibit nanoscale diameters [149], high density [150], large specific surface area [151], and excellent structural controllability [152]. The electrospun fibers produced from such scaffolds possess collagen-like fiber dimensions and hierarchical structures resembling natural tendons, thus presenting promising prospects for applications [153,154].

In principle, electrospun nanofibers can be categorized into two primary groups: natural polymer fibers and synthetic polymer fibers. Nevertheless, in the practical context of tendon repair, it is frequently imperative to blend diverse material sources to fabricate multicomponent polymer composite nanoscale scaffolds to attain functional integration [153,168,169]. By changing the ratio of different material components, one can control the mechanical properties, biodegradability, and bioactivity of electrospun fibers. This enables the macroscopic and microscopic fulfillment of the tensile stress environment in normal tendons and mimicking of the natural niche. For instance, Xue et al. blended silk fibroin (SF) of rigid mechanical properties but low cell affinity with gelatin methacryloy which has relatively lower mechanical strength but stronger cell affinity. Through adjusting the proportion of these components, they achieved a nanofiber scaffold with high mechanical strength and bioactivity. Furthermore, there is a significant enhancement in the production of vascular endothelial growth factors and the expression of tenogenesis in MSCs cultured on this scaffold [170].

However, the unsatisfactory porosity and densely packed fibrous structure of electrospun nanofiber scaffolds may impede cellular growth, proliferation, migration, and infiltration. Additionally, these factors hinder the exchange and diffusion of oxygen and nutrients, impair intracellular transport, and hinder the progression of metabolic processes [155,156]. Therefore, apart from modifying the materials used for biomimetic scaffolds, improving the electrospinning apparatus has gradually emerged as a novel strategy. Wu et al. have developed a novel electrospinning system capable of producing continuous single-axis-aligned nanofiber yarns for fabricating larger-diameter woven fabrics based on PCL (polycaprolactone) nanofiber scaffolds. The study demonstrated that the scaffold with this particular structure exhibits superior porosity, facilitating the adhesion, proliferation, and infiltration of tenocytes [171]. However, its biochemical composition still poses a significant challenge in terms of compatibility with natural tendons. Wu’s team further utilized electrospinning technology to fabricate SF/poly-L-lactic acid (PLLA) nanofiber yarn, complemented by a thermal stretching unit. The resulting nanospun fibers have good mechanical and processing qualities in addition to biological traits. By optimizing the ratio of SF/PLLA, the physical, mechanical, and biological properties can be controlled; it also effectively reduces inflammatory reactions [172].

In summary, the utilization of electrospinning technology in tendon tissue engineering has garnered increasing acknowledgement. Nonetheless, the limitations of low production yield and slow stretching speed pose challenges in achieving large-scale implementation, keeping the technology predominantly confined to the laboratory setting. In addition, it has been reported that the orientation and gravitational forces exerted on the apparatus impacted the fiber diameter and overall spinning efficiency of electrospun products [157]. The road to commercializing electrospinning for clinical applications remains extensive.

## 4. Scaffolds-Based Biomimetic System for Tendon Repair

It is widely acknowledged that different materials possess distinct intrinsic structures and physicochemical characteristics, which have specific effects on the physiological behaviors of cells and the mechanical attributes of the engineered tendon. Biomimetic scaffolds used for tendon repair or regeneration can normally be classified into natural polymer and synthetic polymer scaffolds. A schematic diagram of the simplified tendon regeneration strategies and the biological material classifications are shown in Figure 6. The following sections will detail the development and application of these materials, as well as summarize their advantages and disadvantages in Table 4. Additionally, innovative strategies and novel ideas for tendon repair will also be discussed.

### 4.1. Natural Polymer Scaffolds

Natural biomaterials are available from a wide range of sources and have unique advantages: better cell activity and adhesion than other materials, better biocompatibility and degradation rate, and relatively weak immunogenicity [205,206]. They can be broadly classified into protein-based biomaterials (collagen, gelatin, silk, etc.), glycol-based biomaterials (cellulose, chitosan, alginate, agarose, etc.), glycosaminoglycans (hyaluronic acid, chondroitin sulfate, etc.), and decellularized ECM. The specific details are as follows.

#### 4.1.1. Collagen

Collagen is the main component of the ECM in tendons [207]. Collagen scaffolds have good binding sites for cells and support cell adhesion, migration, and differentiation [122]. Due to its structural similarity to tendons/ligaments [21], it was the first natural polymer used for tendon/ligament reconstruction, and several clinical studies had demonstrated the effectiveness of collagen scaffolds for tendon treatment. Juncosa et al. introduced autologous MSCs into the gel collagen sponge scaffold and found that the linear stiffness and linear modulus of the repaired patellar tendon could reach 75% and 30% of the normal ones [208]. Veronesi et al. wrapped the prepared novel type I collagen scaffold around the sutured Achilles tendon defect in rats. Compared with the control group that only had the defect sutured, the type I collagen content and elastic modulus in the experimental group increased when compared to the control group [209].

To meet the high-intensity stress demand of tendons, Zheng et al. constructed a 3D parallel collagen scaffold using a unidirectional freezing technique and analyzed its effects on tenocyte viability and ECM formation. Their results showed that the scaffold was able to promote the parallel alignment of seeded cells and induce their differentiation toward the tendon lineage (Figure 7) [210]. Sandri et al., likewise, constructed a collagen-BDDGE-elastin (CBE)-based tissue-engineered tendon using a unidirectional freezing technique. They found that this porous shell structural scaffold could direct the cellular arrangement and population with the potential to support and induce in situ tendon regeneration (Figure 8) [211].

In addition, collagen also has the ability to prevent complications such as postoperative adhesions when used for tendon repair [173]. Gelatin is a single-spiral product derived from the hydrolysis of some regions of collagen; it is relatively inexpensive and more stable than collagen. Oryan et al. investigated the applicability of 3D pure bovine gelatin scaffolds for healing in a large tendon defect model in rabbits. The results showed that the rabbit treated with gelatin scaffold exhibited a reduced frequency of peritendinous adhesions and the neotendon presented morphologically more pronounced and larger collagen protofibrils, fibers, and fiber bundles, as well as higher mechanical ultimate load, flexural load, stiffness, maximum stress, and elastic modulus when compared to control group [212]. However, natural collagen-based biomaterials have poor mechanical properties and struggle to withstand higher mechanical stresses [174].

#### 4.1.2. Silk

Silk proteins are natural polymeric fibrous proteins extracted from silk, which have outstanding tensile strength because of their unique fibrous physical form, and largely compensate for the deficiencies of collagen-based materials [175,176]. Altman first explored the potential of a natural 3D scaffold composed of silk for tissue engineering reconstruction of the anterior cruciate ligament, and found that the mechanical properties of this twisted scaffold were similar to those of the human ACL [176,213,214]. Subsequently, Horan et al. investigated the effect of the spinning design on the mechanical properties of silk scaffolds and confirmed that a twisted and cable-like scaffold is more suitable for tendon/ligament tissue engineering which carried high mechanical demands [120]. However, the purification of natural silk protein is costly, the molecular amount is difficult to manage, and mature industrial separation or refinement methods and products are still rare [177]. Thus, researchers have made natural filamentous proteins into filamentous solutions by degumming, solubilization, and concentration to construct regenerated filamentous protein scales in different forms. The main forms currently used for tissue engineering tendons are silk fibers and silk sponges. For centuries, silk fibers have been used as surgical sutures for tendon suturing [215]. Other than surgical sutures, silk fibers with different pore sizes, thicknesses, and mechanical strengths could be made using the knitting method to meet the tendon repair requirements. For example, Fang et al. knitted sericin silk for the repair of Achilles tendon defects in rabbit, and found that the sericin artificial tendon exhibited excellent mechanical strength, withstood the required tendon stress of the organism, and kept the ends of the repaired tendon firmly attached [216].

In addition, different forms of silk could be combined to provide better mechanical support for high-strength tissues such as in tendons/ligaments. Chen et al. recently developed a silk-collagen sponge composite scaffold to improve the disadvantages of cable-like silk that is too dense for cell growth (Figure 9) [217]. They engineered tendons by seeding SCX-overexpressed cells in a silk–collagen sponge scaffold, and showed that the tissue-engineered tendon exhibited more regularly arranged cells and larger collagen fibers under external mechanical stimulation, with better histological scores and mechanical properties than the control group [218]. Similarly, Goh’s team modified an ECM-like protein peptide nanofiber RADA16 on the surface of a filamentous sponge. The in vitro results demonstrated a transient increase in the expression of tendon-related genes, especially the expression of TNC [219]. Afterward, they proposed a coculture method based on a rabbit BMSC/silk proteins scaffold for the repair of osteotendinous articular surfaces, and the results indicated a favorable prognosis [220]. Recently, Hu et al. extracted ligament-derived stem/progenitor cells (LSPCs) from rabbit ligaments for the first time and implanted them in a rabbit ACL reconstruction model in combination with a stromal cell-derived factor 1 (SDF-1) releasing collagen–silk scaffold. At 3 and 6 months postoperatively, this biomimetic silk-protein-based composite graft promoted partial regeneration of tendon and bone tunnels in the ACL and reduced the severity of osteotendinous joint fibrosis (Figure 10) [221]. Shi et al. successfully repaired an osteotendinous interface defect in the ACL by utilizing a composite scaffold of silk proteins with low-crystallinity hydroxyapatite (Figure 11). Presently, the slow degradation [176] rate of silk sericin fibers makes it challenging to match the rate of tissue regeneration, so more studies are needed to improve its application [222].

#### 4.1.3. Spider Silk

Spider silk is a natural material whose main component is fibrous protein that allows the spinning of silk by changing its composition and organization, thus achieving better functionality and adaption to changing environmental conditions. Natural spider silk has excellent mechanical properties, biodegradability, and biocompatibility [123]. Henneck et al. reported that the use of spider silk could stabilize tendon injury [223]. Wendt et al. reported the application of spider silk fibers as an innovative matrix for tissue repair. After sterilization and culturing of fibroblasts for two weeks, keratinocytes were added to the culture to generate a bilayered skin model, suggesting that the spider silk fibers guide the growth of the cells [224]. However, harvesting spider silk from natural sources is not easy and does not provide substantial quantities [178,179]. Recombinant spider silk production enabled the availability of spider silk proteins in larger quantities and of a more consistent quality. This seems to be the best way to produce excellent materials nowadays. It is now gradually gaining attention in the fields of regenerative medicine and tissue engineering. For example, wet-spun fibers made from recombinant spider silk proteins have been used for in vivo studies [225]. The microscopic spider proteins are proteolytically hydrolyzed and spontaneously polymerized into macroscopic fibers, which support the growth of fibroblasts, form emerging capillaries, and are well tolerated [226].

#### 4.1.4. Chitosan and Alginate

Chitosan is a polysaccharide cationic polymer extracted from the sea, which possesses unique properties, such as antibacterial and antioxidant abilities [180]. Zhang et al. inoculated human iPSC-derived MSCs on a chitosan-based microfibrous scaffold, which significantly enhanced the expression of tendon-specific transcription factors and tendon-related genes. Furthermore, the structure and mechanical properties of the repaired tendon treated with this well-aligned fibrous scaffold were superior to the control group [92]. In contrast to chitosan, alginate is a naturally occurring anionic polymeric material with the characteristics of excellent biocompatibility, easy processing, and low cost [227]. Yoon et al. established an alginate scaffold incorporating TGF-β1, which showed a cell proliferation rate of 122.30% and better collagen orientation, continuity, as well as organization at the osteotendinous interface compared to the control group [109]. In addition, as chitosan and alginate are polycationic and polyanionic natural polymeric materials, they could act as crosslinkers to form a composite gel with noncytotoxicity. Camilla et al. implanted an alginate-chitosan scaffold doped with the growth factor rhBMP-2 into a rat rotator cuff injury model. Their results showed that the healing effect and mechanical properties of the scaffold-repaired osteotendinous junction were similar to natural tissue [110]. Tokifumi et al. prepared a model of polyion complex fibers from alginate and chitosan. Their results demonstrated that the alginate-based chitosan hybrid polymer fibers showed much-improved adhesion capacity with fibroblast compared with alginate polymer fiber. Additionally, morphologic studies revealed the dense fiber of the type I collagen produced by the fibroblast in the hybrid polymer fiber dense fibers of type I collagen, suggesting favorable tendon healing [228].

#### 4.1.5. Hyaluronic Acid

Hyaluronic acid (HA) is a high-molecular natural glycosaminoglycan widely distributed in soft tissues. HA has shown lubricating, antiadhesive, anti-inflammatory, as well as high hydration properties, which are important for tendon tissue engineering [229]. Zhao et al. treated flexor tendon tears in dogs with a combination of HA (1%) and lubricin injections, and found a reduction in proximal adhesions that obviously reduced tendon friction to promote sliding [181]. Ozgenel et al. demonstrated that HA could facilitate tendon restoration after repeated injections at the site of flexor tendon injury for 3 months [182]. Furthermore, HA gel could also be used as a carrier for cells, growth factors, and other drugs to improve tissue regeneration. Liang et al. showed that injection of HA-tenocyte complex reduced the inflammatory response after tendon injury [183]. Araque-Monrós et al. fabricated cell carrier microspheres with a PLLA and HA mixture in a 2:1 ratio. After 14 days, the surfaces of the microspheres were completely covered with cells and ECM, and tendon tissue regeneration had significantly improved [230].

#### 4.1.6. Agarose

Agarose, mainly composed of galactose and its derivatives, has self-gelling and reversible thermos-gel properties [184]. These unique characteristics give reason to consider it for tissue engineering applications. For example, Du et al. prepared an agarose/fish gelatin dual network hydrogel using a one-step heating–cooling method, in which the addition of agarose significantly improved the mechanical strength of the composite hydrogel [185]. González et al. developed a fibrin agarose-based hydrogel for the repair of rat Achilles tendons. Their study showed that the tissue regeneration process in the transplanted tendon was vigorous, with relatively organized collagen fiber, cell alignment, and good biomechanics when compared to natural tendon tissue [231]. However, the mechanical properties of agarose resemble those of cartilage tissue, it is more widely used as a cartilage repair material, and the role in tendon repair remains to be studied further.

#### 4.1.7. Cellulose

Cellulose is a polymeric polysaccharide composed of repeatable amino glucose units linked by β-1, 4 glycosidic bonds, which can be categorized into plant cellulose and bacterial cellulose according to the source [232]. Bacterial cellulose has stronger plasticity, biocompatibility, tensile strength, and elastic modulus compared with plant cellulose. In particular, its unique 3D nanofiber structured network resembles tendon tissue, making it an effective raw material for tissue engineering tendons [186,187]. However, the mechanical properties of bacterial cellulose does not meet the mechanical needs of the tendon tissue; it is often used in combination with other materials in tendon tissue engineering. Ramos et al. prepared a hybrid micro-nanofiber using PCL and cellulose acetate (CA), which showed the best adhesion and insulin immobilization of BMSCs when the ratio of PCL:CA was 75:25. Additionally, the expression of tendon markers of the relevant cell phenotype was increased [233]. Nevertheless, the lack of cellulase in most animals makes it almost impossible for bacterial cellulose to complete degradation, which is a fatal flaw and challenge for tissue engineering.

#### 4.1.8. Decellularized Tendon Scaffolds

The complex structural characteristics of tendon ECM make it difficult to restore the original mechanical properties. Thus, mimicking natural ECM to construct novel biological scaffolds remains a great challenge [234]. The advent of decellularization technology has provided a completely new idea for tissue-engineered tendon repair. Compared with traditional natural polymer materials, DECM has a more analogous extracellular structure, low immunogenicity, scalability [126,192], and mechanical properties that resemble natural tissues. More importantly, DECM contains abundant endogenous integrin binding sites, which allows for better morphological, structural, and functional reconstruction of tissues [235]. Cell elution, antigen lysis, and immunogenetic material removal from donor tissues/organs utilizing detergents, enzymes, or physical methods are the most frequently used techniques to obtain decellularized scaffolds [236,237]. Biological scaffolds created using decellularization techniques maintain the natural ECM environmental state, inducing cell migration and regularizing their arrangement patterns [238]. Given that tendon tissue consists of a high matrix component, decellularized biological scaffolds have been widely studied for tendon injury treatment. The materials could be from various origins, such as porcine small intestinal submucosa (SIS), amniotic membrane, and tendon.

The DECM of porcine SIS consists of collagen (about 90%), glycosaminoglycan, fibronectin, and growth factors, which are rich in nutrients to facilitate cell adhesion and proliferation [188]. Chen et al. utilized porcine SIS and type I/III collagen composite as a biological scaffold carrier for autologous tendon seed cells. This scaffold promoted excellent healing and remodeling of rotator cuff tendons at 8 weeks postoperation [239]. Gilbert et al. fabricated a device labeled with 14C of SIS ECM and observed that the tissue, cytoarchitecture, and the vascular distribution of the remodeled SIS ECM was comparable to a normal tendon after 90 days postoperation [240]. However, the applications of SIS DECM in tendon repair are limited due to rapid degradation and poor mechanical properties.

The amniotic membrane is a natural semipermeable membrane on smooth surfaces. It is responsible for cell attachment and proliferation without affecting their immunophenotype and differentiation, which also has antimicrobial, antifibrotic, and anti-inflammatory properties [189]. Amniotic membrane wrapped around partially torn tendon/ligament tissue was demonstrated to reduce inflammation, increase collagen fiber alignment, and improve the mechanical strength of the tendon [190]. For example, Sang et al. reported that decellularized amniotic membranes could accelerate tenocytes proliferation by releasing TGF-β1 and bFGF in vitro, and isolating exogenous adherent tissue while promoting endogenous healing of tendon tissue in vivo [106]. Liu et al. coated PCL nanofibers on the surface of decellularized amniotic membranes using an electrospinning technique to form a multilayer composite membrane, which successfully facilitated the adhesion and proliferation of tenocytes and fibroblasts [191]. However, the bioactive factors in amniotic membranes are hard to preserve for a long time. Additionally, the amniotic membrane is weak in mechanical strength, making it quite difficult to attach to the wound, which is vital in clinical applications [191].

The decellularized tendon matrix (DTM) is one of the most ideal materials for tendon repair, which features excellent biodegradability, high biocompatibility, and low immunogenicity [126,127]. On the premise that the multiunit structure, biomechanical properties, and partial activity factors are well preserved, DTM resembles normal tendon tissue making it suitable for enhancing tendon healing [192,193]. Hiromichi et al. first developed a decellularized multilayered sectioned tendon scaffold loaded with BMSCs, and demonstrated the alignment of cells along tendon collagen fibers and increased expression of tendon-related proteins [241]. Inspired by them, Ning et al. developed a decellularized tendon slice (DTS) scaffold with thickness of 300 μm using Beagle Achilles tendon tissue. They showed that the DTS preserved various ECM microenvironmental signals, including inherent surface topography, biochemical components of tendon ECM, and mechanical behaviors (Figure 12), which contributed to the ability to induce tenogenic differentiation of rat-derived TSPCs and BMSCs in vitro [242]. They further triple stacked the DTS scaffold for full-thickness torn rotator cuff tendon repair in rabbits in a following study. They found that DTS could enhance the adhesion and proliferation of cells, which confirmed the above conclusion as well [243]. To be specific, Tao et al. utilized tandem mass tag labeling proteomics technology to demonstrate that a DTS scaffold could well preserve bioactive components while preventing Achilles tendon adhesions and improving the quality of Achilles tendon repair (Figure 13) [244]. More recently, Xie et al. designed a novel decellularized book-like scaffold (Figure 14) consisting of BMSCs and DTS, which considerably reduced the operational threshold of the conventional cell sheet technique. Moreover, such a novel decellularized “book” tendon scaffold could promote the uniform arrangement and tendon-lineage differentiation of BMSCs [245]. In further studies, decellularized book-like scaffolds loaded with BMSCs were implanted into a rabbit patellar osteotomy model. The in vivo experiments demonstrated an increased expression of patella-patellar tendon-interface-related markers and a significant improvement in the biomechanical properties of the regenerated patellar tendon [246].

Although many studies have shown the positive potential of DECM in tendon tissue engineering, there are still some issues that should be addressed. The underlying mechanisms of DECM from different sources promoting tendon repair remain to be elucidated, especially the outcomes differings from cells. For example, DTS promoted the tenogenic differentiation of TSPCs, whereas dermal-derived DECM had no apparent effect on the differentiation of TSPCs [194]. In addition, dense connective tissues such as tendon and fibrocartilage are highly susceptible to incomplete cell removal, triggering massive inflammatory and fibrotic reactions that limit DECM recellularization [195,196]. Overuse of conventional decellularization reagents (e.g., SDS, Triton-X100, detergents, etc.) impairs the structure and function of bioactive factors in the DECM [247]. Standardizing the decellularization process to ensure the total removal of cells is essential for the application of DECM.

Overall, natural polymer scaffolds hold immense potential for tendon tissue engineering applications, as they show better cell activity and adhesion than other materials, and better biocompatibility and degradation rates. However, their mechanical performance falls to meet the demands of tendon tissue; combination with one or more natural polymers may effectively improve the mechanical properties.

### 4.2. Synthetic Polymer Scaffolds

Synthetic polymeric materials are artificially processed using chemical methods or by the polymerization of different substances. Synthetic polymers have some advantages over natural polymers, including abundant raw material sources, arbitrary modification, and modulation of structure and properties, which are widely adopted in the field of tissue engineering and regenerative medicine. So far, the most common synthetic polymers used in tendon tissue engineering are PGA, polylactide (PLA)/PLLA, PLGA, PCL, etc.

#### 4.2.1. PGA

PGA is a synthetic biodegradable polyester material made of monomeric glycolic acid linked by ester bonds, which provides excellent biocompatibility for the adhesion and proliferation of cells [197]. Liu et al. previously repaired superficial flexor finger tendon defects with a PGA unwoven fiber scaffold seeded with dermal fibroblasts and tenocytes, respectively. Complete degradation of PGA fibers was observed via histology after 14 and 26 weeks. The collagen fibers were aligned in parallel and the mechanical strength of the tendon tissue was well restored in repaired flexor finger tendons [83,248]. However, the degradation rate of PGA in the human body is too fast; it degrades before the nascent tissues are strong enough to bear the in vivo load.

#### 4.2.2. PLA/PLLA

PLA is a polymer produced by the chemical reaction of lactic acid from biological fermentation, which has excellent mechanical properties, biodegradability, compatibility, and a low immune response/cytotoxic reaction [124,125]. Different from PGA, the degradation rate of PLA is comparatively slow. Physical blending or chemical copolymerization of PGA and PLA is emerging as a new option for tissue engineering tendons [249]. For instance, Deng et al. designed a composite core–shell scaffold with an inner layer of PGA nonwoven fibers and an outer layer of PGA/PLA fibers knitted in a 4:2 assembly. A bipolar pattern and a D-periodic structure of collagen fibers similar to natural tendon tissue were observed after 45 weeks of implantation in a rabbit Achilles tendon defect model. The collagen fibers in repaired tendons showed an increasing trend in diameter and tensile strength [250]. Cai et al. fabricated a random and bilayer-arranged silk hibiscus poly(L-lactide-caprolactone) (P(LLA-CL)) nanofiber scaffold using an electrospinning technique. Tendon tissue implanted with the SF/P(LLA-CL) scaffold exhibited better ultimate failure load and stiffness compared to the control group. The microstructure of the tendon-to-bone gradient in the rabbit extra-articular model was enhanced by the above scaffold, inducing bone formation and an increased area of fibrocartilage [251]. Given that growth factors play an important role in tendon healing, Chen et al. constructed a P(LLA-CL)/silk protein nanowire scaffold with GDF-5-induced ADSCs, which showed positive repair potential in rabbit injured tendon tissue [252]. The plasma spraying technique is generally used to prepare surface coatings for tendon grafts. Wu et al. coated the PLA microfiber yarns with PLGA nanofiber, which were obtained using the electrospinning technique using the material surface bioactivation modification technique (Figure 15). The strong mechanical properties of PLA microfibers maintained the structural integrity and load resistance of the tendon. Meanwhile, the nanocoating PLGA fibers endowed the modified PLGA/PLA nanofiber/microfiber hybrid scaffolds with topological cues to guide the behavior of human ADSCs in terms of proliferation, migration, collagen secretion, and tenogenic differentiation [253]. Deepthi et al. also utilized spray coating technology to develop alginate gel-coated and uncoated chitosan-collagen/PLLA scaffolds (Figure 16A), which simulated the natural tendon ECM microenvironment. The study revealed that the porous nature of the hydrogel-coated scaffolds significantly improved cell permeability and adhesion, while enabling them to align uniformly along the fiber direction (Figure 16B,C) to aid effective tendon mechanics [254].

#### 4.2.3. PLGA

PLGA is a linear aliphatic polyester, whose monomers are propylene cross-ester and ethylene glycol [199]. It is widely used in tendon tissue engineering due to its excellent biocompatibility and biodegradability [255]. For example, Sahoo et al. developed a novel, biodegradable nano-microfibrous polymer scaffold by electrospinning PLGA nanofibers onto a knitted PLGA scaffold. This scaffold ensured excellent mechanical strength and better mimicked the nanostructure of natural tendon ECM, which promoted cell attachment, proliferation, and ECM deposition [256]. In their following study, they processed PLGA fibers loaded with bFGF onto the surface of a knitted silk scaffold using electrospinning technology. The release of bFGF could promote the proliferation of mesenchymal progenitor cells and stimulate differentiation toward tendon lineage with enhanced tendon-specific ECM gene and protein expression and collagen production [107]. In addition, Ciardulli et al. fabricated a multi-phase HA/PLGA/fibronectin scaffold using 3D printing, which incorporated knitted HA material, and embedded it in fibronectin hydrogels with PLGA nanocarriers. Their results found an increased expression of both type I collagen and tendon-associated genes in the HA/PLGA/fibronectin scaffold [257]. Similarly, Jiang et al. used 3D technology to combine PLGA scaffolds with collagen fibronectin hydrogels. This composite scaffold effectively promoted growth, proliferation, and tenogenic differentiation of human ADSCs (Figure 17) [131].

#### 4.2.4. PCL

PCL is a linear synthetic biodegradable aliphatic polyester [201]; it has high tensile strength attributed to its unique rheological and viscoelastic properties [202]. Additionally, its ability to mold into different forms makes it an attractive biomaterial used in scaffold development. For example, nanofibers at an average diameter of 1833 ± 369 nm with an elasticity of 6.7 ± 0.4 MPa and a ductility of 587 ± 162% were successfully fabricated based on PCL in a study, which demonstrated an impressive potential for tissue engineering applications [258]. In addition, the advantage of excellent surface area and porosity of PCL allows better metabolic activity and proliferation space for cells [203,204]. For example, Eric et al. prepared polycaprolactone fumarate (PCLF) macroporous composite scaffolds by using PCL and fumarate. Their results showed that cellular bioactivity and the expression of tendon-related genes were increased, and a massive generation of collagen-rich ECM was achieved after the inoculation of human MSCs in 3D PCLF scaffolds, which ultimately accelerated tendon regeneration [259]. Moreover, Li et al. developed an autologous ECM (aECM) scaffold with a hollow and aligned microchannel structure based on PCL microfiber bundle templates. This kind of scaffold stimulated the polarization of cells by generating gradient signals. Targeted surface topography enhanced aECM secretion coupled with the induction of cell migration and alignment towards a high modulus, thus expediting defective Achilles tendon repair in rats [260]. All these studies demonstrated the promise of PCL scaffolds for facilitating the repair of injured tendons in future clinical applications. However, PCL also has some drawbacks including hydrophobicity, a lack of bioactive functional groups, and poor attachment and proliferation of cells [261].

Although, synthetic materials have shown good mechanical strength to meet the acquirements of tendons. The problems regarding poor hydrophilicity, weak adhesion of cells [262], and unsatisfactory histocompatibility are the main problems for them when used as scaffold for tissue engineering [263,264], which should be addressed in future studies.

### 4.3. Innovative Strategies and Advancements in Scaffolds

As mentioned above, both natural and synthetic polymer materials have some drawbacks in application, including poor mechanical strength and a fast degradation rate of natural polymer scaffolds. Synthetic polymer scaffolds have insufficient biological activity, which is not cell friendly and inevitably produces acidic degradation products when degradation in vivo. Therefore, novel-material-based scaffolds for tendon injuries are continuously being explored. For instance, the usage of DECM from pigs and cows could raise religious concerns. Liu et al. developed a novel natural polymeric scaffold using decellularized tilapia fish skin with a smooth outer layer and dense inner structure with a rough surface for tendon tissue engineering. This scaffold exhibited abundant collagen fibers and natural pore structure [265]. In the case of synthetic polymer materials, the main challenge is how to enhance their biological functions. Wang et al. have developed a mechanically tendon-like (0 s UV) QHM polyurethane scaffold (Q: Quadrol, H: Hexamethylene diisocyanate; M: Methacrylic anhydride) immobilized with GDF-7. The scaffold exhibited excellent mechanical strength and provided sufficient biological cues through GDF-7 immobilization, which promoted the sustainable regeneration of tendon-like cells and ECM at the injury site. In addition, composite scaffolds that combine the advantages of natural and synthetic materials have become the hot research topic in tissue engineering [266]. For example, Domingues et al. have utilized cellulose nanocrystals (CNC) as a nano-reinforced filler for coblended matrix PCL/CHT electrospun scaffolds, which fulfill the mechanical stress requirements of tendon tissue engineering and simulate the ECM topographical cues to maintain the morphology and behavior of tenocytes [165]. 

To better achieve biomimetic goals, bioinspired scaffolds that mimic the native tendon niche have also been extensively studied. It is well-recognized that the topographical features of material surfaces play a crucial role in guiding the behaviors of cells [267]. For example, wavy nanofiber scaffolds (WNSs) with parallel fiber alignment, curling characteristics, and nonlinear mechanical properties had been successfully applied in the repair of tendon tissue engineering [268]. This type of wavy scaffold exhibited tendon-like morphological features and had been further validated to effectively enhance the production and assembly of collagen proteins under mechanical stimulation. In addition, biomimetic research is not restricted to the histological level, it also extends to structural biomimetics at the anatomical level. Hwangbo et al. fabricated a uniaxially aligned microtubular collagen scaffold with a lotus-like structure. This microtubular collagen structure with an instructive niche induced highly aligned and efficient myogenic differentiation of myoblasts [269].

Moreover, inspired by the field of bioelectronics, Manus’s team developed piezoelectric collagen-mimetic scaffolds which were composed of aligned nanofibers made from ferroelectric materials such as poly(vinylidene fluoride-co-trifluoroethylene). It had been found that the piezoelectric biodevice could modulate the sensitivity of mechanosensitive ion channels through mechanoelectrical stimulation, thereby facilitating tendon-specific regeneration [270]. With the advancement of technology, more and more new strategies will be developed in the future to enhance the repair and regeneration of tendons.

## 5. Conclusions and Future Perspective

This review highlights the constituents and structural features of tendon tissue and biomimetic scaffolds with natural or synthetic origins for tendon tissue engineering. Biomimetic scaffolds for tendon repair have evolved from passive material containers for transplantation into inducible scaffolds that guide endogenous tissue regeneration. They could mimic the native tissue niche and provide a dynamic biological environment and physicochemical signals to stimulate endogenous tissue repair. Despite the significant developments in the field of tendon tissue engineering, challenges still remain as follows.

It is generally believed that an ideal scaffold should mechanically match that of the native tissue. However, there are still gaps between the mechanical properties of current scaffold materials and natural tendons. The poor mechanical strength of natural materials makes it difficult to maintain the structural integrity of the tendon under a microstress environment. Although this deficiency is effectively improved in mechanically coblended composite scaffolds, the excessive pursuit of matching mechanical properties could result in scaffolds with excessive brittleness and an inability to withstand the applied loads of multidirectional dynamic shear forces. Thus, it is an imperative research direction for the future to design and preferably select suitable scaffold materials to endow the tendon tissue with better mechanical functions.

Regarding the microstructure of tendon tissue, its microfine spatial structure is complex and has a multiscale unit which spans from nanometer to centimeter. Single biomaterials do not cover each scale of tendon tissue; constructing composite scaffolds with different scales to spatially regulate cellular bioactivity and tendon tissue remodeling is essential for tendon tissue engineering. In addition, collagen fibers in normal tendons are mainly aligned along the fiber axis with crimped features. However, current scaffolds for tendon regeneration are mostly only focused on aligned topography, which fail to well mimic the microstructure of tendons. So, to replicate the native tendon ECM properties, both the fiber scale and corrugated superstructure should not be neglected.

More noteworthy is the fact that the process of tendon repair after injury is extremely complicated, which involves the secretion of numerous growth factors. The pathological process after tendon injury is not fully characterized; the growth factors that contribute to tendon repair also needed further investigated. In recent years, the utilization of materiology tools (material surface patterning technology, single molecular layer autonomous mounting technology, construction of flexible, rigid adjustable substrate material technology, etc.) has been demonstrated to help understand the interaction mechanism between the spatial arrangement of material active substances and cellular behaviors at the molecular or nano level, which provides certain ideas for the repair of injured tendons. However, the microenvironment after tendon injury is dynamic; comprehensive understanding of the process and underlying mechanism of tendon healing contributes to designing bioactive scaffolds and identifying the optimal timing of scaffold implantation after injury.

With the rapid development of single-cell technology, the identification of cell subpopulations and developmental trajectory analysis are gradually becoming important in regenerative medicine. The underlying theory is to reveal the functions and properties of potential repair cells through single-cell analysis. Therefore, the use of single-cell technology to understand the functions of different cell subpopulations is beneficial to develop new scaffolds for directing the differentiation of different subpopulation. Lastly, most studies on current scaffolds are limited to small animals in laboratories. Standardization of the large animal models, evaluation systems, stringent quality testing, and multicenter clinical certification are necessary to promote the translation of tendon scaffolds in clinic.

## Figures and Tables

**Figure 1 biomimetics-08-00246-f001:**
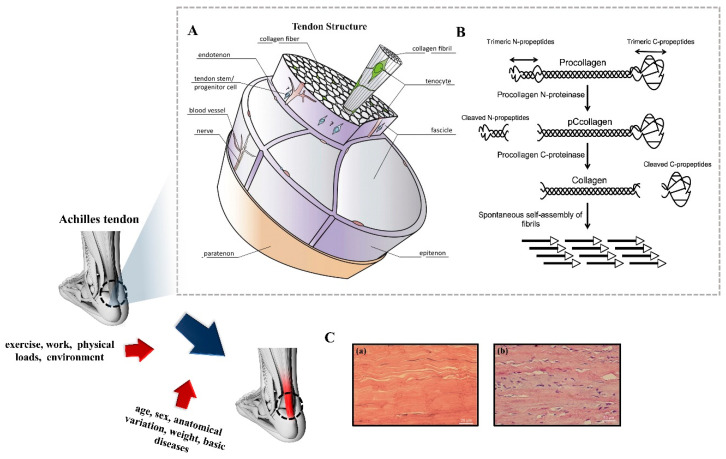
Hierarchical arrangement of basic tendon structures, pathological changes, and mechanisms of restoration after injury: (**A**) A schematic drawing of basic tendon structure. Reproduced with permission from Ref. [32]. Copyright 2014, Elsevier B.V., Amsterdam, The Netherlands. (**B**) Schematic representation of collagen fibril formation by cleavage of procollagen. Reproduced with permission from Ref. [35]. Copyright 2017, *International Journal of Experimental Pathology*. (**C**) Histological differences between normal and tendinosis tendon tissue. The normal tendon shows organized collagen fibers and a sparse amount of tenocytes, tightly packed between the collagen bundles (**a**). In tendinosis (**b**), the tendon structure becomes disorganized, the tenocytes change morphology and proliferate. Reproduced with permission from Ref. [39]. Copyright 2015, Christensen et al.

**Figure 2 biomimetics-08-00246-f002:**
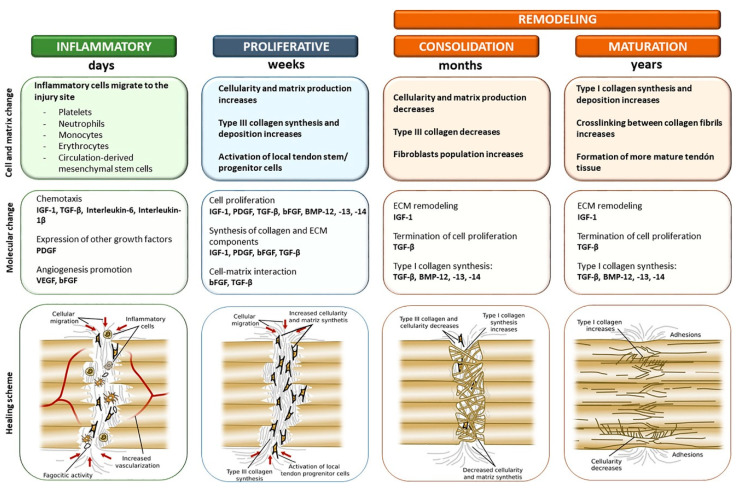
Main changes produced during the different phases of tendon regeneration: inflammatory phase, proliferative phase, and remodeling phase. Reproduced with permission from Ref. [46]. Copyright 2021, Elsevier B.V.

**Figure 3 biomimetics-08-00246-f003:**
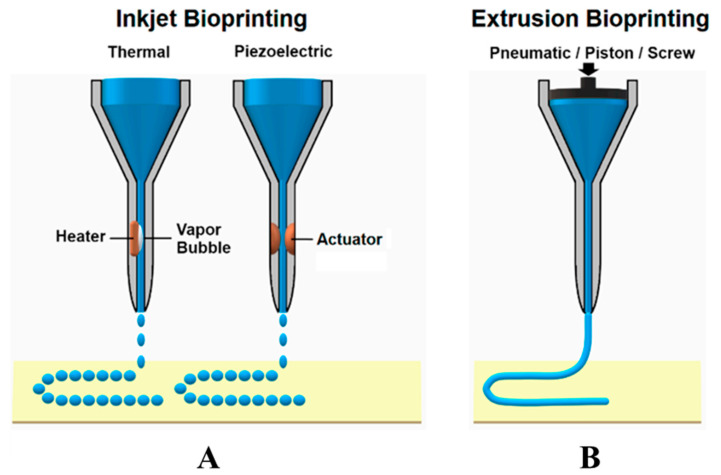
Common bioprinting techniques: (**A**) Inkjet bioprinting uses an electric heater or piezoelectric actuator to create a pressure pulse that propels the bioink droplet onto the substrates. (**B**) Extrusion bioprinting utilizes a pneumatic or piston or screw-based pressure to extrude the bioink through a micronozzle in the form of a continuous filament. Reproduced with permission from Ref. [139]. Copyright 2020 by the authors. Licensee MDPI, Basel, Switzerland.

**Figure 4 biomimetics-08-00246-f004:**
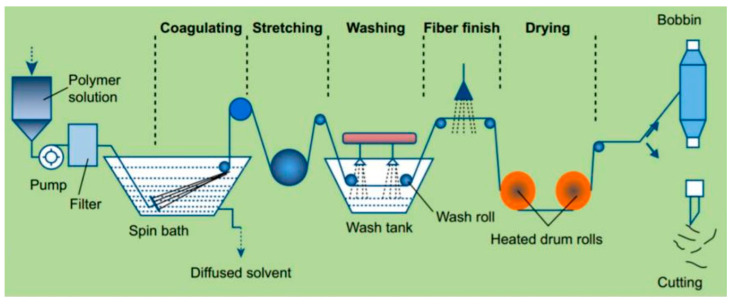
Schematic illustration of the wet-spinning process for yarn preparation. Reproduced with permission from Ref. [164]. Copyright 2021 by the authors. Licensee MDPI, Basel, Switzerland.

**Figure 5 biomimetics-08-00246-f005:**
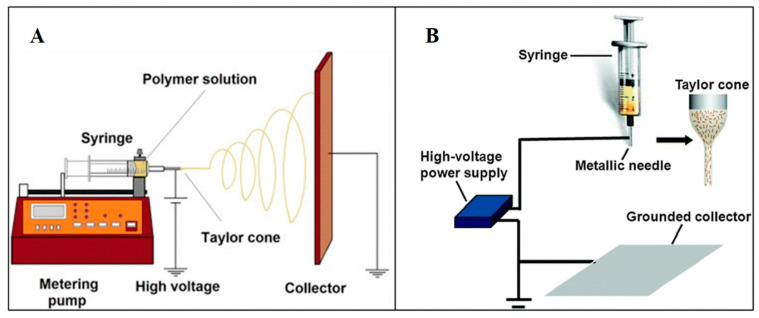
Schematic diagram of setup of electrospinning apparatus: (**A**) Horizontal setup; (**B**) Vertical setup. Reproduced with permission from Ref. [167]. Copyright 2019, Springer Nature Switzerland AG.

**Figure 6 biomimetics-08-00246-f006:**
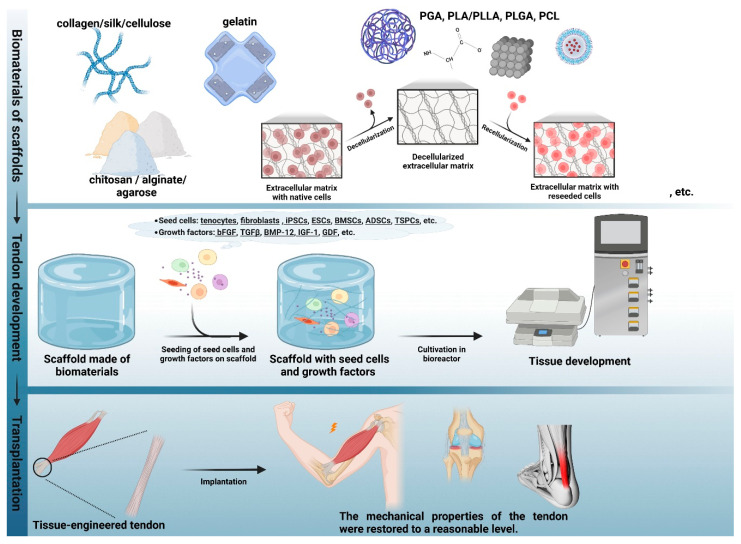
Diagram of typical elements in tendon tissue engineering and synthesis patterns of biomimetic scaffolds.

**Figure 7 biomimetics-08-00246-f007:**
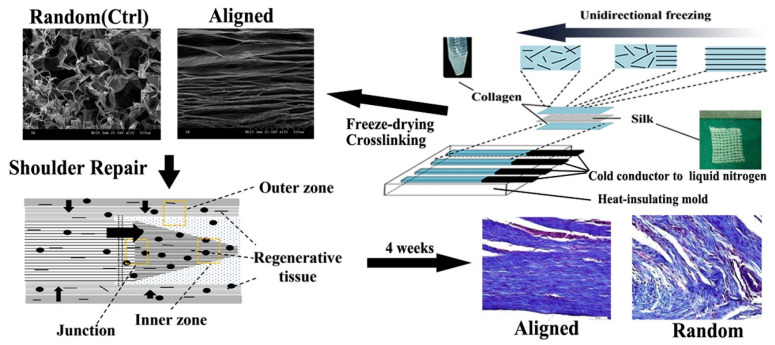
SEM images of 3D-oriented collagen scaffolds obtained using the unidirectional freezing technique and control scaffolds, as well as a schematic representation of local tissue regeneration after scaffold implantation and Masson staining of them. Reproduced with permission from Ref. [210]. Copyright 2017, Elsevier Ltd., Amsterdam, The Netherlands.

**Figure 8 biomimetics-08-00246-f008:**
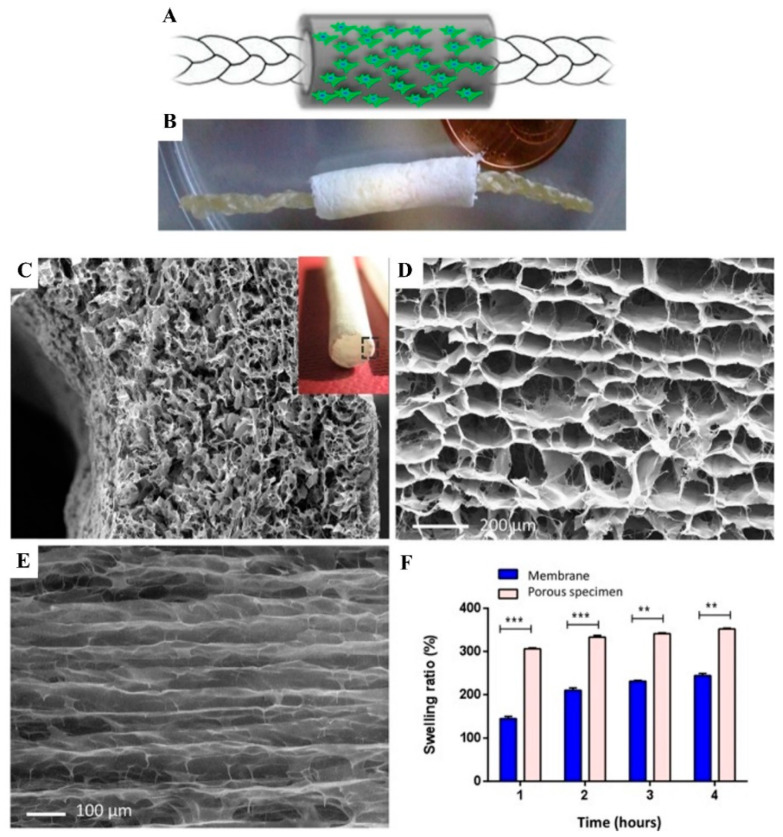
Schematic representation (**A**) and macroscopic appearance (**B**) of the core-shell scaffold architecture. Micro- and macro- (insert) structure of the shell scaffold (**C**). Axial view of the shell pore microstructure, endowed with an anisotropic axial porosity (**D**). Longitudinal view showing preferential pore orientation along the axis of the tube (**E**). Swelling behavior of the porous shell specimen in PBS solution, compared with the CBE membrane (300 μm thick): *** *p* ≤ 0.0001; ** *p* ≤ 0.001 (**F**). Reproduced with permission from Ref. [211]. Copyright 2016, Sandri et al.

**Figure 9 biomimetics-08-00246-f009:**
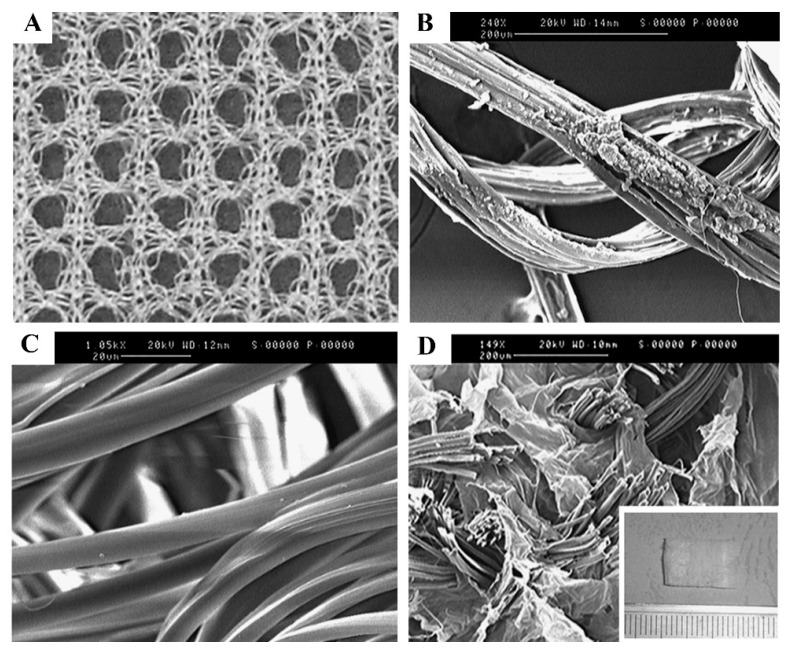
Knitted silk scaffold (**A**) and SEM (**B**,**C**) images of the changes in surface texture of a single fibroin fiber prepared using different procedures. The gross and SEM images show combined silk scaffolds showing collagen sponge formed in the openings of the knitted silk scaffold (**D**). Reproduced with permission from Ref. [217]. Copyright 2008, Elsevier Ltd.

**Figure 10 biomimetics-08-00246-f010:**
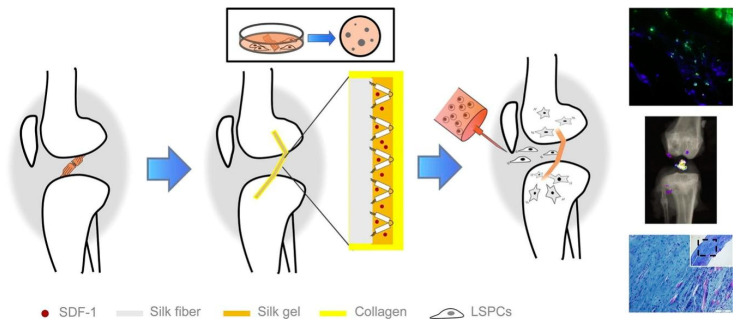
Schematic diagram for in vivo tracking of intra-articular injected CFDA-labeled LSPCs. Reproduced with permission from Ref. [221]. Copyright 2018, Elsevier Ltd.

**Figure 11 biomimetics-08-00246-f011:**
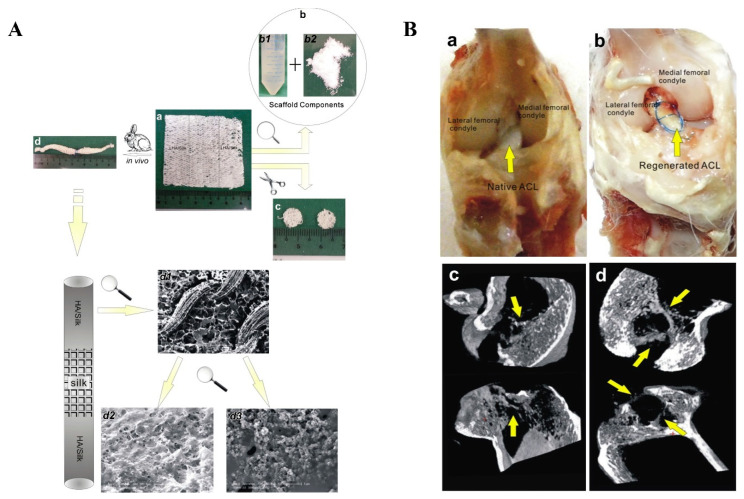
(**A**) Diagrammatic sketch of experimental design: gross view of knitted silk mesh with both ends modified by LHA (**a**); and the components of scaffolds are consisted of silk solution (**b1**) and LHA (**b2**); The scaffolds are trimmed in disk shape for in vitro tests (**c**); the whole knitted silk mesh is rolled up for implantation (**d**); detailed structure of the rolled-up scaffolds: the scaffold having a highly porous structure (**d1**), and decoration of CHA and LHA contributes to distinct surface morphology variation (**d2**,**d3**). (**B**) Native ACL (**a**) and regenerated ACL after 4 months of implantation (**b**); and m-CT images of implant-bone junction in femoral and tibial bone tunnels after 2 and 4 months implantation (**c**,**d**). The yellow arrows point to new bone in the μ-CT images. Reproduced with permission from Ref. [222]. Copyright 2013, Elsevier Ltd.

**Figure 12 biomimetics-08-00246-f012:**
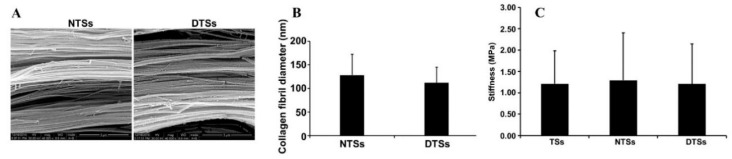
SEM characterization for the surface topography of NTSs and DTSs. Scale bar ¼ 3 mm (**A**). The collagen fibril diameter of NTSs and DTSs (*p* > 0.05) (**B**). Stiffness of TSs, NTSs, and DTSs as determined by AFM (*p* > 0.05) (**C**). Reproduced with permission from Ref. [242]. Copyright 2015, Elsevier Ltd.

**Figure 13 biomimetics-08-00246-f013:**
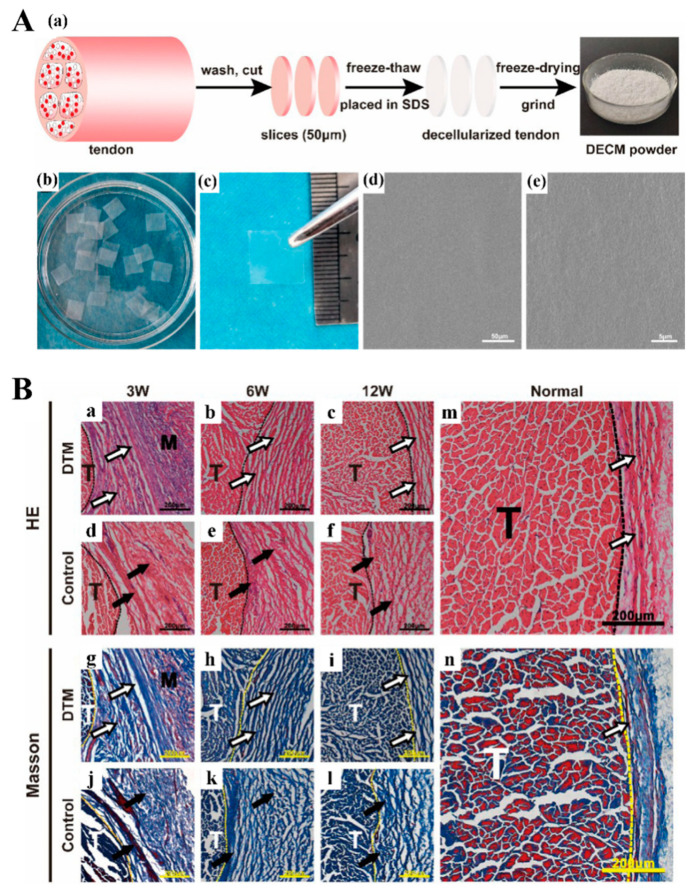
(**A**) Preparation and characterization of tendon matrix membranes. Preparation from tendon to DECM powder (**a**). Representative macroscopic images of wet DTM (**b**) and dry DTM (**c**). The surface microstructures of DTM are observed by SEM at magnifications of × 1000 (**d**) and × 8000 (**e**). (**B**) Evaluation of tendon matrix membranes in the repair of a rabbit Achilles tendon. H&E staining (**a**–**f**) and Masson staining (**g**–**l**) of tissues around tendon after Achilles tendon repair. DTM group, suture with DTM; control group, suture without DTM. H&E staining (**m**) and Masson staining (**n**) of normal rabbit tendon without surgery Reproduced with permission from Ref. [244]. Copyright 2021, Elsevier Ltd.

**Figure 14 biomimetics-08-00246-f014:**
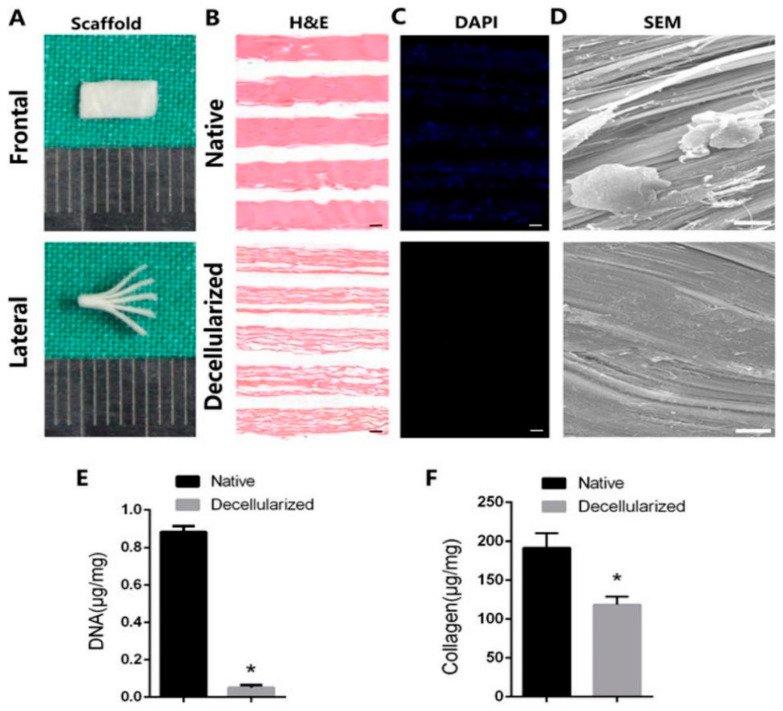
Preparation of the book-shaped DTM scaffold (**A**). H&E staining of the native and decellularized “book” tendon scaffold, scale bars = 100 μm (**B**). DAPI staining of the native and decellularized “book” tendon scaffold, scale bars = 100 μm (**C**). SEM of the native and decellularized “book” tendon scaffold, scale bars = 20 μm (**D**). DNA and collagen contents of the native and decellularized “book” tendon scaffold. * Significant difference between native and decellularized group (*p* < 0.05) (**E**,**F**). Reproduced with permission from Ref. [245]. Copyright 2019, Orthopaedic Research Society.

**Figure 15 biomimetics-08-00246-f015:**
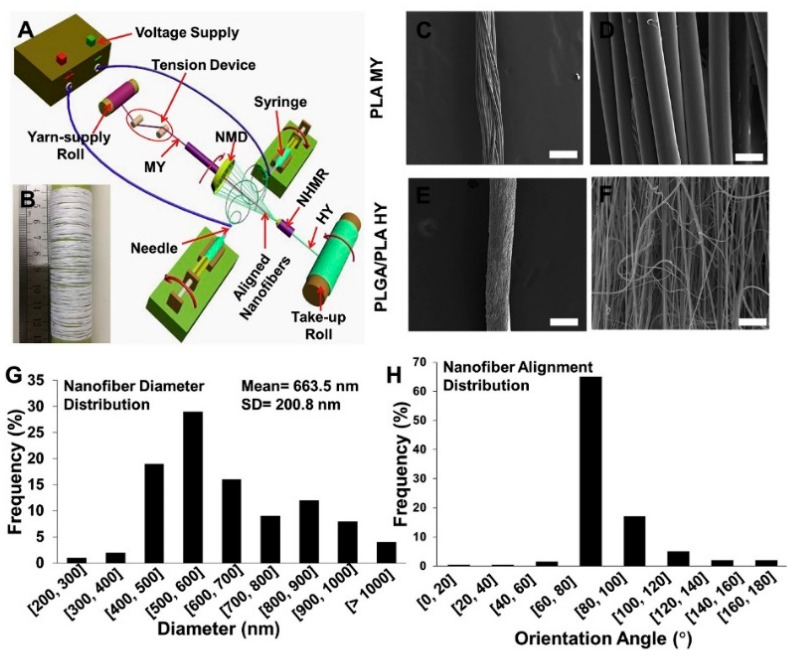
Fabrication of PLGA/PLA nanofiber/microfiber HY. (**A**) Schematic illustration of the modified electrospinning system for coating electrospun PLGA nanofibers on the surface of PLA MY to generate PLGA/PLA HY. (**B**) Photograph of a PLGA/PLA HY package with fabrication and electrospinning for 4 h. (**C**,**D**) SEM images of the original PLA MY; (**E**,**F**) SEM images of the obtained PLGA/PLA HY. Scale bars: 200 μm for (**C**,**E**); 20 μm for (**D**,**F**). (**G**) Fiber diameter distribution of PLGA nanofibers on the surface of PLGA/PLA HY. (**H**) Orientation of the angular distribution measurement of the PLGA nanofibers on the surface of PLGA/PLA HY. Reproduced with permission from Ref. [253]. Copyright 2019, Elsevier B.V.

**Figure 16 biomimetics-08-00246-f016:**
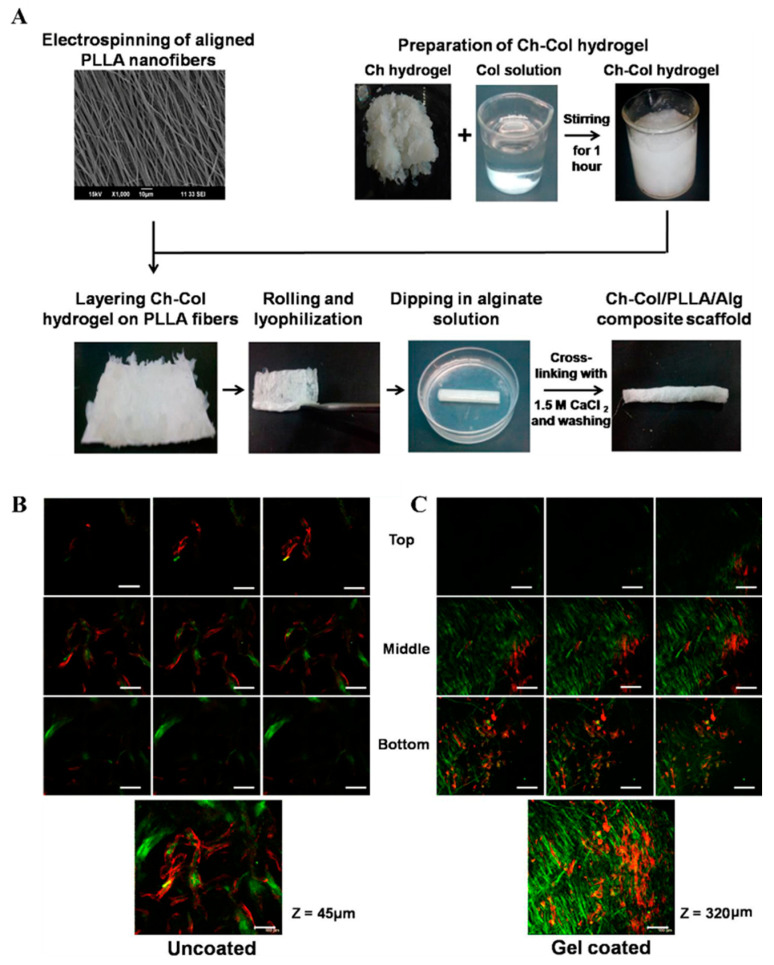
(**A**) Pictorial representation of fabrication of Ch-Col/PLLA/Alg composite construct. Confocal images for tenocyte cell infiltration on uncoated scaffold (**B**) and gel-coated scaffold (**C**) after 7 days. Scale bar denotes 100 μm. Reproduced with permission from Ref. [254]. Copyright 2016, Elsevier Ltd.

**Figure 17 biomimetics-08-00246-f017:**
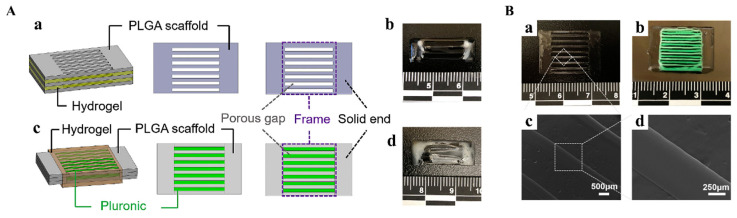
(**A**) Schematics of two PLGA scaffold models with collagen-fibrin hydrogels. (**a**) Schematic illustration of the separate layerby-layer structure. Three layers of PLGA scaffolds sandwiched with two layers of collagen-fibrin hydrogels in between injected by pipette. (**b**) Final separate layerby-layer structure of PLGA scaffolds with collagen-fibrin hydrogels. The PLGA scaffolds were cut in half equally before the collagen-fibrin hydrogels were injected. (**c**) Schematic illustration of the tri-layered structure. Three layers of PLGA and Pluronic F127 were printed as a whole structure. Pluronic F127 was washed out in cold water after printing. Collagen-fibrin hydrogels were injected by pipette and wrapped around the three-layer PLGA scaffolds. (**d**) Final tri-layered structure of PLGA scaffolds with collagen-fibrin hydrogels. The Pluronic F127 was washed out. The PLGA scaffolds were cut in half equally before the collagen-fibrin hydrogels were injected (**B**) Two types of 3D printed scaffolds. (**a**) 3D-printed one-layer PLGA scaffold for the separate layer-by-layer model. (**b**) 3D-printed tri-layered scaffold model using PLGA and Pluronic F127. Pluronic F127 was dyed with green food coloring. (**c**,**d**) SEM images of one-layer PLGA scaffold. Ref. [131]. Copyright 2020, the authors. Publishing services by Elsevier B.V. on behalf of KeAi Communications Co., Ltd., Beijing, China.

**Table 1 biomimetics-08-00246-t001:** Advantages and disadvantages of seed cells for tendon repair.

Seed Cells	Advantages	Disadvantages	Ref.
Tenocytes	Major cell type in the tendon Direct repair of tendon	Limited sources and quantities	[60,61,62]
Fibroblasts	Rapid migration and proliferation Easy to obtain from skin	Limited lifespan Scar formation	[63,64,65]
ESCs	High potential for differentiation Unlimited proliferative ability	Ethical concern Potentially tumorigenicity	[66,67,68]
iPSCs	No ethical concern High potential for differentiation Unlimited proliferation capacity	Genotype shift in the transfection process Potentially tumorigenicity Epigenetic variation	[69,70,71]
BMSCs ADSCs	Easily available Autologous transplantation No ethical concern	Loss of phenotype Limited quantities Senescence during passage Ectopic bone formation	[72,73,74,75]
TSPCs	Excellent capability for tenogenesis	Easy loss of tenocyte phenotype Limited sources and quantities	[76,77,78,79]

**Table 2 biomimetics-08-00246-t002:** Summary of the major targets and effects of growth factors in tendon repair.

Growth Factors		Main Roles	Effects	Ref.
bFGF			Promoted endogenous healing and barrier exogenous healing of tendon.	[106]
			Stimulated initial proliferation and subsequent tenogeneic differentiation of the BMSCs enhancing their collagen production.	[107]
		Bound to cell membrane receptors	Regulated proliferation of cells and promoted the expression of type III collagen.	[108]
TGF-β	TGF-β1	Key inducer of tenogenesis Mediated RhoA/ROCK signaling and mechanotransduction pathway	Induced transdifferentiation of narrow fibroblasts into tenocytes to enhance tendon repair.	[84]
			Promoted collagen synthesis, angiogenesis, and matrix protein regulation.	[109]
			Regulated tenocytes proliferation and differentiation and promoted collagen I production.	[106]
	TGF-β3	Key inducer of tenogenesis Paracrine pathway	Reflected key roles for regeneration compared with adult tendon.	[90]
			Promoted the expression of SCX, ELN, and TNC to reduce scar formation.	[95,96]
BMP	BMP-2		Activated downstream genes that induce stem cells into tenocytes.	[90]
			Induced the formation of collagen type I and resulted in a more native-like osteotendinous junction with better biomechanical properties.	[110]
	BMP-12	Triggered robust phosphorylation of Smad1/5/8 that conveyed by type I receptors ALK2/3/6	Increased the expression of the tendon marker scleraxis and tenomodulin at both mRNA and protein levels.	[111]
IGF-1		Activated PI3K/protein kinase B and ERK pathways Matrix production	Regulated collagen synthesis and tenocytes proliferation.	[82]
GDF	GDF-5		Induced ectopic bone and cartilage both in vivo and in vitro and caused the migration of host progenitors.	[90]
	GDF-6		Increased expression of tendon membrane proteins to induce tenogenic differentiation of the BMSCs.	[112]
	GDF-8		Induced tenogenic differentiation of the pluripotent stem cells.	[113]

**Table 3 biomimetics-08-00246-t003:** Common techniques for scaffold fabrication.

Fabrication t\Techniques		Advantages	Disadvantages	Ref.
3D bioprinting	Inkjet printing	High-speed printing Strong controllability Continuous printing High resolution and accuracy Good compatibility	Unable to print high-viscosity materials and high-density cells Mechanical or thermal damage to cells	[131,132,133,134,135,136,137,138]
	Extrusion-based bioprinting	Sedimentation of high-density cells	Poor cell viability	[139,140,141]
Wet-spinning		Dissolving biomacromolecules Retaining the hydration Artificially adjustable fiber diameter, porosity, and pore size	Poor size stability Limited by fiber properties Micrometer diameter	[142,143,144,145,146,147,148]
Electrospinning		Nanoscale diameters High density Large specific surface area Excellent structural controllability Collagen-like fiber dimensions and hierarchical structures similar to natural tendons	Unsatisfactory porosity Low production yield Slow stretching speed Be impacted by gravity	[149,150,151,152,153,154,155,156,157]

**Table 4 biomimetics-08-00246-t004:** Examples of scaffold biomaterials in tendon tissue engineering.

Source	Biomaterial	Advantages	Disadvantages	Ref.
Natural	Collagen	Biocompatibility Biodegradability Main component of tendon ECM	Poor mechanical strength Fast degradation	[21,122,173,174]
	Silk	Flexibility Excellent tensile strength High processability	Limited cell adhesion Difficult to manage molecular weight Slow degradation rate	[175,176,177]
	Spider silk	Superior strainability Excellent mechanical properties Biodegradability Biocompatibility	Limited natural production Hardening easily when exposed to air	[123,178,179]
	Chitosan	Antibacterial Antioxidant	Low mechanical properties	[180]
	Alginate	Biocompatibility High processability Low cost	Low mechanical properties Limited cell adhesion	[128]
	Hyaluronic acid	Lubricant Less adhesion Low inflammatory Direct injection	Low mechanical properties Fast degradation with natural form	[181,182,183]
	Agarose	Reversibility Excellent mechanical properties	Less studies related to tendons	[184,185]
	Cellulose	High plasticity Biocompatibility Excellent mechanical strength 3D structure	Low mechanical strength Hard to degrade in vivo ∙	[186,187]
	SIS	Rich nutrients	Rapid degradation Poor mechanical properties	[188]
	Amniotic membrane	Rich nutrients Biocompatibility Biodegradability Antibacterial Antifibrotic Low inflammatory	Difficult to store for a long time Low mechanical strength Easily torn	[189,190,191]
	DTM	Biocompatibility Biodegradability Low immunogenicity Structure close to natural tendons	Influenced by sources Low purification	[126,127,192,193,194,195,196]
Synthetic	PGA	Biocompatibility No toxic products	Rapid degradation	[197]
	PLA/PLLA	Excellent mechanical properties Biodegradability Biocompatibility Low inflammatory, immunogenicity, and cytotoxic reactions	Low degradation High brittleness	[124,125,198]
	PLGA	Biocompatibility Biodegradability Excellent mechanical properties	Poor hydrophilicity Poor absorption	[199,200]
	PCL	Biocompatibility Biodegradability Excellent tensile strength Excellent surface area and porosity	Hydrophobicity Less bioactive functional groups Poor attachment and proliferation of cells	[201,202,203,204]

## Data Availability

Not applicable.

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
