# Peer review of "Biomimetic Scaffolds for Tendon Tissue Regeneration"

_biomimetics, 2023, doi:10.3390/biomimetics8020246_

Round 1

Reviewer 1 Report

Overall this review needs serious editing to be considered for publication. While the sections of the review are well organized, the rationale for each section lacks organization and lacks clarity. In addition, paragraph structure needs to be reviewed. Here are some critical issues that need to be addressed. 

1. Clear and concise writing. In the very first sentence, "The tendon is a type of connective tissue that resembles a cord and connects bone to muscle." Instead, consider revising to, "Tendon is a connective tissue with chord-like architecture that attaches muscle to bone." 

2. Edit the entire manuscript for clarity. In the second sentence, "It allows for the storage and release of enough efficiency through stretch or contraction of the tendon..." This makes no sense; tendon absolutely does not contract.

3. Way too much time is spent discussing tendon injuries in the second paragraph. This should be its own section in the review. 

4. Another example of weak organization is the third paragraph where extracellular matrix proteins are all of a sudden introduced in a paragraph that started off with vascular distribution - no background information provided at all. There should be at least one sentence that first introduces the role of extracellular matrix in tendon. 

These are all microcosms of the overall nature of this paper. Weakly organized, weak rationale, and no clarity. 

Reviewer 2 Report

The submitted manuscript represents a very comprehensive review of the current state of tissue engineering as it pertains to tendon repair/regeneration.  It is the opinion of this reviewer that the authors have effectively assembled a wealth of relevant knowledge associated with the subject matter and have further organized that content in an easy to navigate flow.  The content of this manuscript will contribute meaningfully to the field of tissue engineering.  Though the manuscript contains multiple graphics adopted from previously published work, this does work to strengthen the overall work and avoids wall of text that might impact readability.  It also appears that all appropriate copyrights were obtained for these figure reproductions.  The included tables serve well to concisely summarize content from the referenced studies.  Based on these points, it is the opinion of this reviewer that the submitted manuscript be accepted in its present form following editorial screening for minor spelling/grammatical/formatting errors.

Reviewer 3 Report

I thank the authors for this very good work.
This is a high-level study that has been rigorously conducted.
The conclusions of the work are extremely interesting and will allow better management of patients with tendon injuries. I have no comments or corrections to make. I see no objection to the publication of this article

Reviewer 4 Report

Although the work is a concise review of the utilization of various natural and synthetic polymers on tendon tissue engineering, the main issue of biomimetic scaffold fabrication techniques has not been addressed properly. Biomimetic strategies for tendon/ligament regeneration are not highlighted. Restructuring of the manuscript is necessary.

Literature should be updated, only 37 out of 245 references are after 2020

Current strategies and novel ideas should be discussed in a separate chapter

Round 2

Reviewer 4 Report

The paper can be accepted in its present form.

Author Response

Thank you for the positive comments and affirmation of our work.